# Reliable Image Quality Evaluation
# and Mitigation of Quality Bias in Generative Models

## Abstract

Discrepancies in generation quality across demographic groups pose a substantial and critical challenge in image generative models. However, the Fréchet Inception Distance (FID) score, which is widely used as an image quality evaluation metric for generative models, introduces unintended bias when assessing quality across sensitive attributes. This undermines the reliability of the evaluation procedure. This paper addresses this limitation by introducing the Difference in Quality Assessment (DQA) score, a novel approach that quantifies the reliability of existing evaluation metrics, e.g. FID. DQA assesses discrepancies in evaluated quality across demographic groups under strictly controlled conditions to effectively gauge metric reliability. Our findings reveal that traditional quality evaluation metrics can yield biased assessments across groups due to inappropriate reference set selection and inherent biases in image encoder in FID. Furthermore, we propose DQA-Guidance within diffusion model sampling to reduce quality disparities across groups. Experimental results demonstrate the utility of the DQA score in identifying biased evaluation metrics and present effective strategies to mitigate these biases. This work contributes to the development of reliable and fair evaluation metrics for generative models and provides actionable methods to address quality disparities in image generation across groups.

## 1. Introduction

In recent years, image generative models such as Generative Adversarial Networks (GANs) (Goodfellow et al., 2020), Denoising Diffusion Probabilistic Models (DDPMs) (Ho

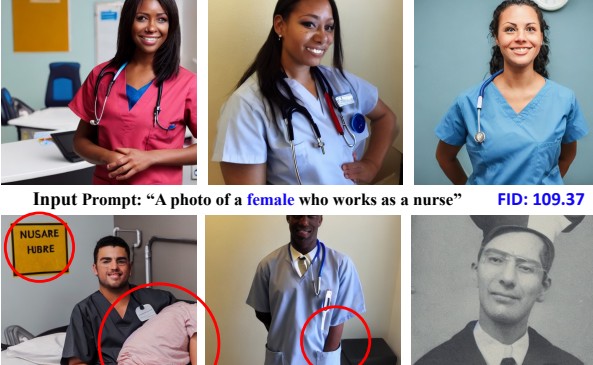

*Figure 1.* Using the same prompt template and seed, a generative model may produce varying image quality across different demographic groups, e.g., generating higher-quality nurse images for females while producing obscured objects, distorted limbs, or grayscale images for males.

et al., 2020), and text-to-image generation (Ramesh et al., 2021; Rombach et al., 2022) systems have brought bias concerns to the forefront of generative modeling. While substantial research has focused on distributional fairness to ensure balanced sample generation across sensitive attributes (Choi et al., 2024; Shen et al.; Li et al., 2023; Parihar et al., 2024; Jung et al., 2024), the fairness in generation **quality** across demographic groups remains an equally critical yet underexplored issue. For example, Fig. 1 demonstrates the existing bias in generation quality by producing better quality of image for certain demographic group.

Furthermore, in the classification task, text-to-image generative models can be used as data augmentation tools to improve classifier performance (Kim et al., 2024). However, if the quality of generated images is inconsistent across demographic groups, it can negatively impact classification performance for certain groups, exacerbating fairness issues in prediction and introducing biases in decision-making. We empirically demonstrate in Appendix A that discrepancies in image generation quality can adversely affect real-world applications, e.g. medical imaging (Garcea et al., 2023), particularly in classification performance and fairness (Lar-

[1]Anonymous Institution, Anonymous City, Anonymous Region, Anonymous Country. Correspondence to: Anonymous Author <anon.email@domain.com>.

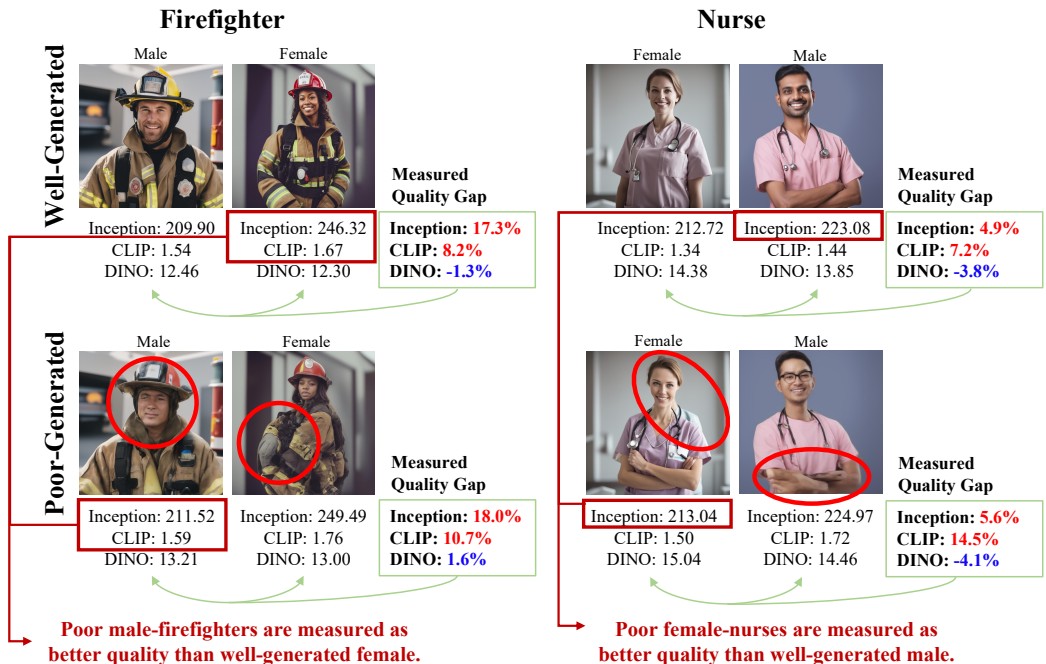

*Figure 2.* Using the same distance metric (Fréchet Distance, smaller is better), we compare image quality across varying professions and genders, with each set consisting of 1,000 images. Each image set is carefully controlled to include both well-generated and poorly-generated images. We evaluate three image encoders: InceptionV3 (FID), CLIP, and DINO. A biased encoder in quality evaluation leads to two forms of unreliable measurement. First, InceptionV3 and CLIP exhibit significant measurement gaps across demographic groups for images of the same quality, whereas DINO shows relatively smaller discrepancies. Second, InceptionV3 and CLIP misleadingly assess poor-quality images as having better quality, while DINO more accurately reflects true quality assessments.

razabal et al., 2020). We also show that achieving fair quality in generated images can lead to improved outcomes, underscoring the necessity of addressing this issue.

In response, recent studies (Perera & Patel, 2023; Naik & Nushi, 2023) have highlighted quality discrepancies in generative models related to gender-profession biases, relying on the Fréchet Inception Distance (FID) (Heusel et al., 2017) to assess the quality of generated images. However, our analysis reveals that FID is unreliable for evaluating fairness in image quality for two reasons.

First, FID is sensitive to the selection of reference dataset due to distinct group distributions. As demonstrated in our synthetic data analysis in Sec. 3 and Fig. 3, the reference should be chosen group-specific manner. Choosing combined dataset as reference for FID not only leads to inaccurate quality evaluations for each group but also misidentifies the direction of bias, making FID an unreliable metric for detecting fairness issues in generative models observed in (Perera & Patel, 2023; Naik & Nushi, 2023).

Secondly, even with group-specific evaluation, traditional encoders can remain unreliable due to inherent biases in image encoders, which may produce inconsistent representations for images of similar quality across demographic

groups. For example, as shown in Fig. 2, biased encoders such as InceptionV3 and CLIP yield unreliable evaluation results, misassessing certain demographic groups as having better image quality.

We identify that this inconsistency arises from the biased representations produced by the encoder. To validate this issue, we use a t-SNE (Van der Maaten & Hinton, 2008) plot of embeddings from a biased encoder, shown in Fig. 4 (b). The plot reveals a clear gender-based separation despite similar image quality, highlighting the encoder's failure to reliably evaluate quality discrepancies across demographic groups. Further details are provided in Sec. 3.2.

In summary, although quality bias exists in generative models, the commonly used evaluation metric, FID, and potential alternatives leveraging different backbone networks (Jayasumana et al., 2024) are not reliable for assessing this bias. This raises the following key questions:

**Q1: Which image encoder for evaluation metric can reliably assess quality bias, and how can it be quantified?**
**Q2: What strategies can effectively mitigate quality bias in generative models?**

We summarize the contributions of this paper by addressing these two questions.

To address the first question, we introduce a novel score, the *Difference in Quality Assessment* (DQA), which serves as a **reliability score** for assessing the reliability of evaluation metrics' fairness across demographic groups. DQA quantifies whether an encoder introduces bias, by measuring discrepancies in evaluation results across demographic groups based on strictly controlled test dataset. An encoder with a lower DQA value is interpreted as more reliable and suitable for group-specific quality assessments to be used as an evaluation metric for image quality.

DQA can identify the most reliable pre-trained foundational models in quality evaluation in Sec. 4, supporting fairness and reliability in future generative model applications for downstream tasks. Additionally, in Appendix A, we validate DQA's effectiveness by adopting a classification task with data augmentation using a text-to-image generation, showing that DQA-guided data augmentation improves fairness in classification performance. Although DQA is not specifically designed to improve classification fairness, these results highlight its effectiveness as a reliability metric for achieving quality fairness in generated dataset.

Furthermore, to address the second question, we propose a DQA-based regularization method, **DQA-Guidance** for diffusion models' sampling stage, which enhances both quality fairness and overall generation quality without retraining the diffusion model, as discussed in Sec. 5.

## 2. Related Work

### 2.1. Generated Image Quality Assessment

FID is a widely used metric for assessing the quality of generated images by measuring the Wasserstein-2 distance (Vaserstein, 1969) between embeddings of synthetic and real images extracted by the InceptionV3 (Szegedy et al., 2016). This embedding-based distance measurement has thus become standard in generative model research (Sauer et al., 2025; Koh et al., 2024; Wang et al., 2024; Bansal et al., 2024). To enhance representational richness and relax distributional assumptions, MMD with the CLIP encoder (Radford et al., 2021) has been proposed (Jayasumana et al., 2024). While prior studies (Bińkowski et al., 2018; Chong & Forsyth, 2020; Jain et al., 2023) have highlighted the unreliability of evaluation metrics under finite or imbalanced sample conditions, the reliability of these metrics from a fairness perspective remains largely unexplored.

### 2.2. Fairness in Generative Models

Many studies have explored fairness in generative models but have primarily focused on addressing distributional bias, aiming to achieve an equal number of generated samples across demographic groups from a neutral prompt such as fine-tuning the entire model (Choi et al., 2024; Shen

et al.), utilizing a pretrained classifier (Li et al., 2023; Parihar et al., 2024), and manipulating intermediate embeddings (Jung et al., 2024). Some works concentrate on new metric evaluating such biases (Cho et al., 2023; Sathe et al., 2024).

In contrast, beyond distributional bias, Perera & Patel (2023) and Naik & Nushi (2023) highlighted that quality bias in generated images across demographic groups, particularly in associating certain careers with specific genders. However, methods for mitigating quality bias have not been presented in the literature. We are the first to propose guiding the diffusion model's sampling stage to ensure fairness in image quality.

## 3. Bias in Image Quality Assessment for Generative Models

Recent studies have highlighted concerns about quality bias in generated images (Perera & Patel, 2023; Naik & Nushi, 2023). To evaluate the quality of generated images and quantify this bias, the Fréchet Inception Distance (FID) (Heusel et al., 2017) is widely used as a metric for assessing the similarity between the distributions of real and generated images. FID calculates the statistical distance between embeddings extracted from the InceptionV3 model (Szegedy et al., 2016) for both generated images and a reference dataset (Brack et al., 2023; Feng et al., 2022; Saharia et al., 2022; Podell et al., 2023). However, as discussed in Sec. 1, relying on FID for quality evaluation presents significant limitations.

### 3.1. Selection of Reference Dataset

Firstly, the measurement method should be group-specific to accurately capture differences across demographic groups. To formalize, let $D(\cdot, \cdot)$ denote a distance measurement such as Maximum Mean Discrepancy (MMD) (Radford et al., 2021) or Fréchet Distance (FD), and let $f$ represent an image encoder. Define two demographic groups $A$ and $B$, with corresponding reference datasets, $A_{\text{ref}}$ and $B_{\text{ref}}$, and generated datasets, $A_{\text{gen}}$ and $B_{\text{gen}}$. The combined reference and generated datasets are given by $\mathcal{I}_{\text{ref}} = A_{\text{ref}} \cup B_{\text{ref}}$ and $\mathcal{I}_{\text{gen}} = A_{\text{gen}} \cup B_{\text{gen}}$. In FID, $D$ represents FD while $f$ is typically the InceptionV3 model (Szegedy et al., 2016). In the quality bias literature (Perera & Patel, 2023; Naik & Nushi, 2023), the generation quality of each group is calculated by $D(f(A_{\text{gen}}), f(\mathcal{I}_{\text{ref}}))$ and $D(f(B_{\text{gen}}), f(\mathcal{I}_{\text{ref}}))$ for groups $A$ and $B$, respectively, while the bias measurement is given by $D(f(A_{\text{gen}}), f(\mathcal{I}_{\text{ref}})) - D(f(B_{\text{gen}}), f(\mathcal{I}_{\text{ref}}))$. Here, the magnitude represents the degree of bias, while the sign indicates its direction.

However, as demonstrated in our synthetic data analysis in Fig. 3, using a unified reference dataset can mask or amplify biases, potentially leading to unfair assessments of image quality across different groups. In this figure, blue

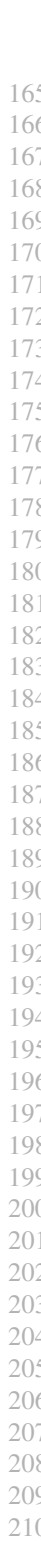

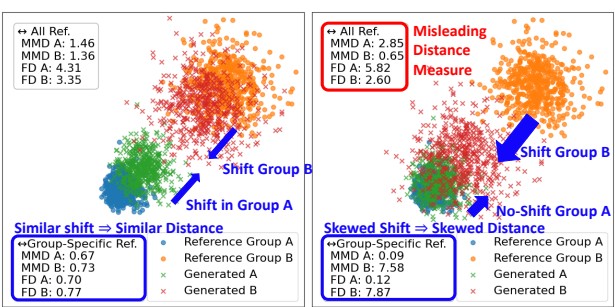

(a) Example of Fair Image Encoder    (b) Example of Unfair Image Encoder

*Figure 3.* Illustration of quality bias in evaluation metrics using distance measures such as Maximum Mean Discrepancy (MMD) and Fréchet Distance (FD). The left figure depicts a fair scenario where generated data embeddings for both groups exhibit the same distribution shift, while the right figure shows an unfair scenario, with embeddings for one generated group skewed towards the other. Using group-specific references (e.g., $A_{gen} \leftrightarrow A_{ref}$) more accurately captures distribution shifts compared to an all-reference approach (e.g., $A_{gen} \leftrightarrow A_{ref} \cup B_{ref}$), which can produce misleading values in cases of biased image encoders. Thus, group-specific distance measures more accurately evaluate the quality under the biased representation.

and orange points represent reference embeddings for two demographic groups, while green and red points denote generated embeddings for each group. Fig. 3 (a) depicts a scenario where the embeddings of the generated data are similarly out-of-distribution from their respective reference datasets, suggesting a fair assessment. In contrast, Fig. 3 (b) shows a scenario where the generated data embeddings for one group are skewed toward the other group's reference data, indicating potential quality bias. According to Fig. 3 (b), the quality evaluation results for group $B$ should be worse (higher) than for group $A$. However, when using the combined reference set, as denoted as "All Ref.", the measured distances indicate $D(f(A_{\text{gen}}), f(\mathcal{I}_{\text{ref}})) \gg D(f(B_{\text{gen}}), f(\mathcal{I}_{\text{ref}}))$, which is misleading. In contrast, in Fig.3 (a), using group-specific references yields $D(f(A_{\text{gen}}), f(A_{\text{ref}})) \ll D(f(B_{\text{gen}}), f(B_{\text{ref}}))$, providing an accurate evaluation. Thus, the quality bias evaluation should be $D(f(A_{\text{gen}}), f(A_{\text{ref}})) - D(f(B_{\text{gen}}), f(B_{\text{ref}}))$, in a group-specific manner, rather than $D(f(A_{\text{gen}}), f(\mathcal{I}_{\text{ref}})) - D(f(B_{\text{gen}}), f(\mathcal{I}_{\text{ref}}))$.

### 3.2. Bias in Image Encoder Used in Evaluation

Secondly, when discrepancies in group-specific quality evaluations are observed, it remains unclear whether these differences stem from actual variations in image quality or from biases inherent in the image encoder. A biased encoder can distort embeddings, impacting the interpretation of image quality across groups and leading to skewed evaluation results, as observed in Fig. 2. We illustrate

this issue in Fig. 4 (a), and verify this in Fig. 4 (b) using t-SNE plot. In Fig. 4 (b), although well-generated images are correctly located closer to each reference, a poorly generated image of a "male nurse" may be embedded closer to the "female nurse" reference due to encoder bias, rather than reflecting its true quality. Conversely, a similarly poor-quality image of a "female nurse" remains within the in-distribution region of the "female nurse" reference, indicating inconsistency in quality evaluation across demographic groups. This leads to inaccuracies in both quality assessment and quality bias evaluation, such that $|D(f(A_{\text{gen}}), f(A_{\text{ref}})) - D(f(B_{\text{gen}}), f(B_{\text{ref}}))| \gg 0$, even though $TrueQuality(A_{\text{gen}}) \approx TrueQuality(B_{\text{gen}})$.

Given these limitations, it is crucial to identify evaluation metrics that can reliably distinguish between distribution shifts caused by actual quality discrepancies and those resulting from biases in the image encoder. By employing group-specific measurement and introducing a reliability score for evaluation metrics using controlled, same-quality images, we can better understand the sources of quality bias and improve the fairness and accuracy of image quality assessments across different demographic groups.

## 4. Reliability of Evaluation Metric for Generated Image Quality

In this section, we introduce a novel method to assess the reliability of evaluation metrics for generated image quality, focusing primarily on metrics that measure the distributional distance between generated and reference datasets. This emphasis arises from concerns that biased image encoders might handle poor-quality images inconsistently across sensitive groups, even when distances are calculated in a group-specific manner, as discussed in Sec. 3.1.

### 4.1. Difference in Quality Assessment

We consider two generated datasets, $A_{\text{gen}}$ and $B_{\text{gen}}$, each containing images of comparable quality and equal quantity. In our experiments, we use MMD as a distance metric $D(\cdot, \cdot)$ instead of FD due to its efficiency and freedom from distributional assumptions (Jayasumana et al., 2024).

*Difference in Quality Assessment* (DQA) aims to identify bias in the evaluation metric $D(f(\cdot), f(\cdot))$. Recalling the combined reference and generated datasets as $\mathcal{I}_{\text{ref}} = A_{\text{ref}} \cup B_{\text{ref}}$ and $\mathcal{I}_{\text{gen}} = A_{\text{gen}} \cup B_{\text{gen}}$, DQA is formulated as:

$$\text{DQA} = \frac{\left| D(f(A_{\text{gen}}), f(A_{\text{ref}})) - D(f(B_{\text{gen}}), f(B_{\text{ref}})) \right|}{D(f(\mathcal{I}_{\text{gen}}), f(\mathcal{I}_{\text{ref}}))}$$

(1)

By employing group-specific distance measurements, Eq. (1) isolates the bias inherent in the encoder by comparing the embeddings of generated images with consistent

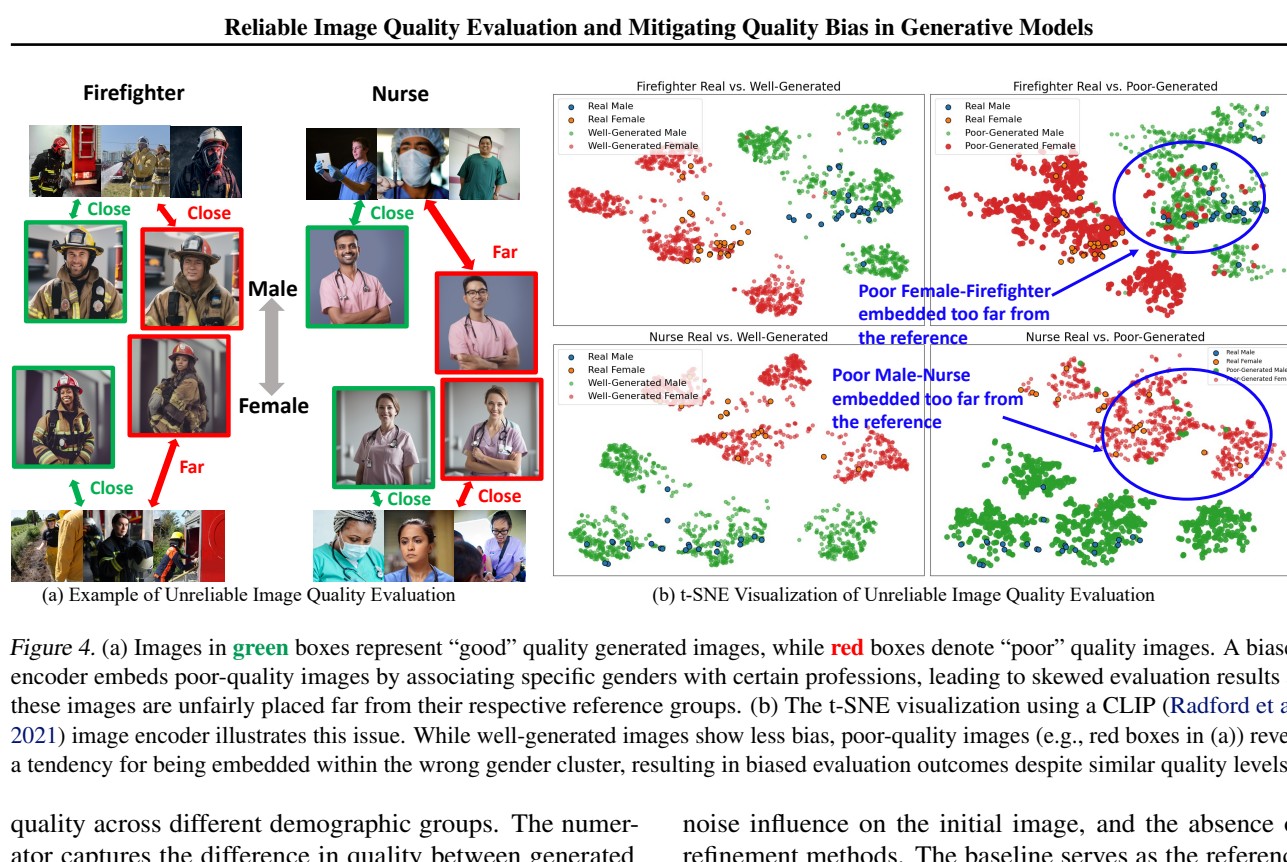

(a) Example of Unreliable Image Quality Evaluation  (b) t-SNE Visualization of Unreliable Image Quality Evaluation

*Figure 4.* (a) Images in **green** boxes represent "good" quality generated images, while **red** boxes denote "poor" quality images. A biased encoder embeds poor-quality images by associating specific genders with certain professions, leading to skewed evaluation results as these images are unfairly placed far from their respective reference groups. (b) The t-SNE visualization using a CLIP (Radford et al., 2021) image encoder illustrates this issue. While well-generated images show less bias, poor-quality images (e.g., red boxes in (a)) reveal a tendency for being embedded within the wrong gender cluster, resulting in biased evaluation outcomes despite similar quality levels.

quality across different demographic groups. The numerator captures the difference in quality between generated data for groups $A$ and $B$ relative to their respective reference sets. A large numerator implies significant quality disparity between groups, whereas a small or zero value suggests the encoder treats both groups equally. The denominator captures the global generation quality by measuring the distance between the combined reference and generated datasets. A smaller denominator value indicates generated data closely matches the reference set, while a larger value signifies deviation. Hence, DQA quantifies the relative quality discrepancy between groups compared to the overall distribution shift in generation. A low DQA suggests fair treatment of both groups by the encoder, while a high DQA indicates significant bias. Therefore, DQA serves as a **reliability score** for quantifying bias in image encoders.

### 4.2. Constructing the Evaluation Dataset for DQA

To effectively apply the DQA score for finding reliable image encoders in practice, it is essential to construct controlled reference and generated datasets. To assess the reliability of image encoders, we construct a dataset with six different versions, ranging from well-generated to poorly generated sets, capturing realistic scenarios encountered in text-to-image generation of human images using Stable Diffusion XL (SDXL) (Podell et al., 2023). Following the recommended settings from (Lui et al., 2024) as our baseline, we degrade image quality in various ways by adjusting hyperparameters. The scenarios include the baseline, weak guidance, reduced sampling steps in diffusion, increased

noise influence on the initial image, and the absence of refinement methods. The baseline serves as the reference dataset, while the other scenarios represent controlled generated datasets. For each image seed, we prepare datasets under all six scenarios. We generate 250 images for each combination of profession, gender, and race, resulting in 20,000 images per scenario (10 professions, 2 genders, and 4 races). This ensures that each attribute has the same number of reference images, avoiding inaccuracies caused by imbalanced attribute distributions (Jain et al., 2023). Detailed descriptions of each degradation, along with the professions and races used, are provided in Appendix C, and visualizations are presented in Fig. 5 (a).

### 4.3. DQA for Multiple Attributes (e.g., Race)

Let Eq.(1) be denoted as $\text{DQA}(A_{gen}, B_{gen}; f)$ for groups $A$ and $B$ given encoder $f$. Let $\mathcal{G} = \{G_1, \cdots, G_n\}$ represent the set of $n$ groups. We aggregate pairwise DQA across all combinations to provide a comprehensive measure of fairness in image quality assessment across multiple attributes.

$$\text{AvgDQA}(\mathcal{G}) = \frac{1}{\binom{n}{2}} \sum_{1 \leq i < j \leq n} \text{DQA}(G_i, G_j; f), \quad (2)$$

### 4.4. Reliability Analysis for Pre-trained Image Encoders

To assess the reliability of image encoders in evaluating generated image quality fairly across demographic groups, we apply the DQA score to various pre-trained models, considering differences in architecture, training dataset, and

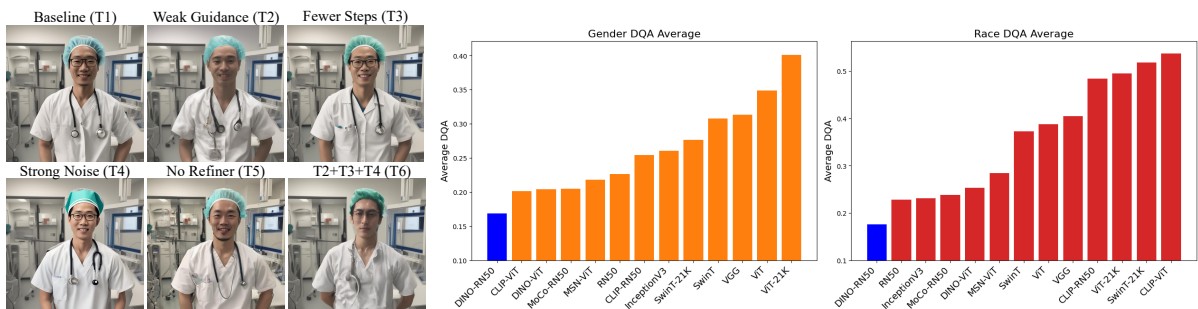

(a) Example of  Dataset for DQA with Controlled Degradation          (b) DQA of Various Models for Controlled Evaluation Dataset

*Figure 5.* (a) Examples of generated images under controlled degradation scenarios. The figure illustrates samples from both the well-generated baseline (reference, T1) and the intentionally degraded cases (T2 - T6), where image quality is systematically reduced by adjusting specific hyperparameters. This controlled degradation enables effective measurement of the DQA score to assess the reliability of an image encoder. (b) Across all pre-trained encoders and various degradation in generated images, DINO-RN50 achieves the lowest DQA in average, indicating it is the most reliable encoder for evaluating the quality of generated images.

training scheme. In this analysis, we calculate the average DQA score across all degradation types.

We evaluate models including InceptionV3, VGG (Simonyan & Zisserman, 2014), ResNet-50 (RN50), ViT-B/16 (Dosovitskiy, 2020), and Swin Transformer (Liu et al., 2021), all trained on the ImageNet-1K (IN-1K) (Deng et al., 2009) dataset using supervised learning. We also compare models trained on IN-1K and ImageNet-21K (IN-21K) (Ridnik et al., 2021) for ViT-B/16 and Swin Transformer architectures to examine the effect of training dataset size. Additionally, we explore different training schemes by evaluating models trained with self-supervised methods like MoCo-RN50 (He et al., 2020), MSN-ViT (Assran et al., 2022), and DINO (Caron et al., 2021) and CLIP using both RN50 and ViT-B/16 architectures.

**Impact of Training Scheme on DQA.** Our results, summarized in Fig. 5 (b) indicates that self-supervised models using the RN50 architecture, particularly DINO-RN50 and MoCo-RN50, achieve the lower DQA scores in general compared to supervised models. This suggests that the combination of self-supervised learning and the RN50 architecture effectively reduces bias, leading to fairer embeddings across demographic groups. We analyze this as self-supervised models learn representations without explicit labels, which helps them avoid inheriting biases tied to label information.

**Impact of Backbone Network on DQA.** In contrast, self-supervised models using the ViT architecture, such as DINO-ViT and MSN-ViT, exhibit slightly higher DQA scores, implying that RN50 may be better suited for learning unbiased representations in self-supervised settings. We analyze the architectural differences between convolutional neural networks (CNNs) (Schmidhuber, 2015) and Transformers (Vaswani, 2017). RN50, as a CNN, incorporates locality and spatial patterns through its convolutional layers.

This structure allows CNNs to capture both local and global image features, making them more robust to distortions in the image (Tuli et al., 2021). In contrast, Transformer-based models rely on self-attention mechanisms that process images as sequences of tokens, without the same spatial locality constraints (Tuli et al., 2021). The token-based approach enables the model to capture complex global dependencies, but it may also make it more sensitive to specific variations in distorted images (Guo et al., 2023), resulting in larger discrepancies between reference and generated datasets.

**Impact of Training Dataset on DQA.** We also examine the effect of training dataset size by comparing models trained on IN-1K and IN-21K for both ViT-B/16 and Swin Transformer. The results show that models trained on the larger dataset, IN-21K, actually exhibit higher DQA scores compared to their IN-1K counterparts. This suggests that increasing the dataset size alone does not necessarily improve fairness in the encoder's representations. Similarly, models like CLIP, despite being trained on large-scale image-text datasets, show higher DQA scores especially in racial bias, indicating that large-scale multimodal training does not necessarily guarantee fairness in embeddings.

### 4.5. Validity of DQA

To validate the effectiveness of DQA for quality assessment, we apply it to data augmentation in a medical image classification task. As detailed in Appendix A, datasets generated by text-to-image models for medical images can be used for data augmentation but often exacerbate fairness issues due to quality bias in the generative model, resulting in significant performance gaps across demographic groups in classification. Leveraging a reliable image encoder, we construct both fair and unfair generated datasets based on their DQA scores as detailed in Algorithm 1. Fair dataset

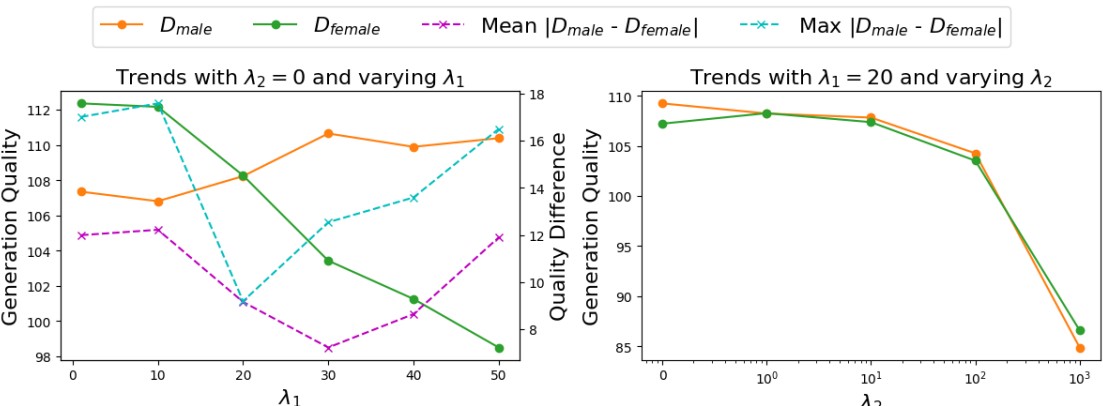

*Figure 6.* Experimental results for generation quality and quality disparities with DQA-Guidance with Stable DIffusion. The left plot shows the impact of $\lambda_1$ on generation quality for each demographic group (lower values indicate better quality) and displays the average and maximum quality gap across all disease classes (lower values indicate reduced disparity). The right plot illustrates the effect of $\lambda_2$ on overall generation quality. Here, $\lambda_1 = 0$ denotes no DQA-Guidance, while higher $\lambda_1$ values reflect a stronger influence of DQA-Guidance. DQA-Guidance effectively enhances generation quality and reduces quality disparities across demographic groups.

enhances classification fairness when used for augmentation, whereas unfair dataset exacerbates disparities. This demonstrates DQA's ability to identify reliable image encoders and its practical utility in enabling DQA-based data augmentation. These findings underscore the benefit of DQA in generative models for classification applications, as further elaborated in Appendix A.

## 5. Mitigating Quality Bias in Diffusion Models

DQA serves not only as a reliability indicator for the evaluation metric but can also act as an energy function in generative models to regularize equal image quality across demographic groups. Specifically, we employ guided diffusion (Liu et al., 2022; Epstein et al., 2023; Bansal et al., 2023) during sampling in diffusion models rather than training a model from scratch. By interpreting DQA as an energy function, we can incorporate its gradient into the diffusion sampling process to mitigate bias in image generation. This approach leverages the principles of energy-based guidance, where gradients of an energy function are used to steer the generation process toward desired outcomes without modifying the pre-trained model parameters.

### 5.1. DQA-Guidance for Diffusion

In our context, the DQA score quantifies relative discrepancies in image quality assessments across demographic groups. By computing the gradient of DQA with respect to latent variables $z_t$ at each diffusion timestep, we obtain the latent direction that reduces this discrepancy. Incorporating this gradient into noise prediction adjusts the sampling trajectory to favor samples that minimize quality differences across groups.

Assume we identify a reliable image encoder $f^*$ for evaluating generated image quality. Let $g$ be the base generative model that samples from latent variable $z_t^A$ and $z_t^B$ for each group. We apply DQA-Guidance in diffusion modeling by taking the gradient of DQA with respect to $z_t = [z_t^A; z_t^B]$:

$$\tilde{\epsilon}_\theta(z_t) = \epsilon_\theta(z_t) + \sigma_t \lambda_1 \nabla_{z_t} \mathrm{DQA}(g(z_t^A), g(z_t^B); f^*), \quad (3)$$

where $\epsilon_\theta(z_t)$ is the estimated noise, $\theta$ represents the pre-trained weights of the diffusion model, $\sigma_t$ scales the gradient term according to the noise level at timestep $t$, and $\lambda_1$ is a hyperparameter controlling the strength of the DQA-Guidance in diffusion process.

Since reducing DQA could unintentionally increase the denominator of DQA (representing the overall quality), we introduce an additional term to ensure that both the numerator and denominator are minimized. Specifically, we add the gradient of the denominator of DQA, the overall distributional distance between generated and reference datasets $D\big(f^*(\mathcal{I}_\mathrm{gen}), f^*(\mathcal{I}_\mathrm{ref})\big)$, as a regularizer to improve quality:

$$\tilde{\epsilon}_\theta(z_t) = \epsilon_\theta(z_t) + \sigma_t \nabla_{z_t} \Big( \lambda_1 \mathrm{DQA}(g(z_t^A), g(z_t^B); f^*)$$
$$+ \lambda_2 D\big(f^*(\mathcal{I}_\mathrm{gen}), f^*(\mathcal{I}_\mathrm{ref})\big) \Big), \quad (4)$$

where $\lambda_2$ is a hyperparameter balancing the influence of the quality regularizer. By incorporating both terms, we ensure that the model not only reduces the quality bias but also maintains high overall image quality.

Thus, by treating DQA as an energy function and integrating its gradient into the diffusion sampling process, we effectively guide the generation toward reducing the quality disparity while preserving the fidelity of the generated images.

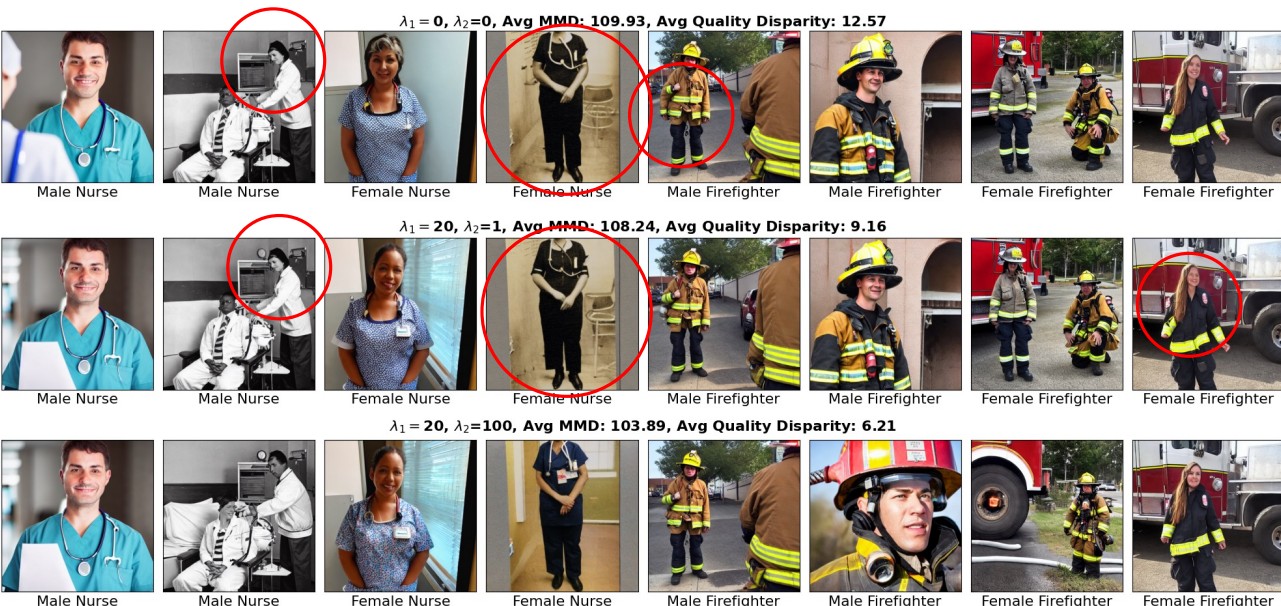

*Figure 7.* Qualitative results of DQA-Guidance for human image generation. The examples demonstrate improvements in artifact reduction, color correction, and texture and background refinement (marked as red circle). These enhancements illustrate the impact of DQA-Guidance in balancing quality across demographic groups.

## 5.2. Experimental Details for DQA-Guidance

To verify the effectiveness of DQA-Guidance in mitigating quality bias, we conduct experiments generating human images using Stable Diffusion. We utilize the well-generated (Baseline) dataset introduced in Appendix C as a reference set to maintain consistency in quality and context across demographic groups when computing DQA during the diffusion process. To evaluate the impact of DQA-Guidance, we apply it to Stable Diffusion (Rombach et al., 2022). In this setup, images generated by the state-of-the-art model (SDXL) (Podell et al., 2023) are used as the reference set, and DQA-Guidance helps to mitigate quality disparities while enhancing overall image quality in the diffusion model. The extension of DQA-Guidance for medical image generation with ImageGen (Saharia et al., 2022) is introduced in Appendix F.

## 5.3. Result Analysis for DQA-Guidance

Fig. 6 demonstrates the clear impact of DQA-Guidance on image generation. Compared to the baseline ($\lambda_1 = 0$), increasing $\lambda_1$ effectively reduces quality disparities in generated images while substantially improving overall image quality, especially $\lambda_1 = 20$ and $\lambda_1 = 30$. However, setting $\lambda_1$ too high introduces excessive noise, leading to a decline in image quality. These findings suggest that DQA not only provides a reliable measure for evaluating fairness but also serves as an effective regularizer, enhancing fairness

in image generation when applied as guidance in diffusion models. Additionally, larger values of $\lambda_2$ intuitively contribute to improved generation quality, as demonstrated in Fig. 6 (b). Qualitative results of DQA-Guidance are presented in Fig. 7, demonstrating improvements in average quality (denoted as Avg MMD) while also reducing the quality gap (denoted as Avg Quality Disparity).

## 6. Conclusion

In this paper, we address the underexplored issue of quality disparities in image generation models and introduce the Difference in Quality Assessment (DQA) score as a novel approach for assessing the reliability of evaluation metrics in measuring generated image's quality. Through extensive analysis, we reveal that commonly used metrics, such as FID, can introduce unintended biases, resulting in misinterpretation of quality discrepancies due to the use of combined reference sets and model sensitivity to specific demographic features. DQA mitigates these issues by guiding users in identifying reliable image encoders, thus providing a more accurate and dependable measure of quality fairness in generative tasks. We further enhance the utility of DQA through DQA-Guidance in diffusion models, demonstrating that this approach effectively reduces quality disparities across groups while preserving high image fidelity. These findings establish a robust framework for advancing fairness in generative models, setting a more reliable standard for quality assessment across diverse demographic groups.

## Impact Statement

This work addresses critical gaps in fairness and reliability in image quality evaluation for generative models, a pressing concern in applications such as healthcare and social media. The proposed Difference in Quality Assessment (DQA) approach provides a novel approach to identifying biases in existing evaluation methods that highlights the challenges posed by pre-trained encoders, which may carry inherent biases. This underscores the need for ongoing efforts to refine foundational models.

The DQA-Guidance framework further demonstrates how quality fairness can be integrated into the generation process without retraining, promoting more inclusive and accessible applications of generative AI. These contributions are particularly impactful in fields like medical imaging, where biased models can exacerbate health disparities, and in domains where equitable representation across demographics is critical.

Overall, this research advances the development of equitable and reliable generative AI, fostering responsible innovation in technologies that promote societal fairness and support decision-making.

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

# A. Impact of Quality Bias in Generative Models in Downstream Task and Validity of DQA

## A.1. Negative Impact of Quality Bias in Generative Models

Unfairness in generated image quality across demographic groups poses a critical issue in generative modeling. Generative models, especially those trained on uncurated datasets, often produce images of systematically lower quality for specific demographic groups, such as those defined by gender, race, or age. This quality discrepancy not only undermines visual representation fairness but also risks reinforcing biases when these generated images are used for data augmentation in training pipelines, potentially transferring such biases into downstream models. Addressing this issue requires robust strategies to ensure consistent image quality across all demographic attributes.

To highlight the practical implications of quality bias, we conduct a classification task with a ResNet-50 model (He et al., 2016) using chest X-ray images from the Chest X-ray dataset (Wang et al., 2017), a dataset known to exhibit fairness issues, as evidenced by differing AUC scores across demographic groups (Larrazabal et al., 2020). To enhance classifier's performance, a user might employ text-to-medical-image generation models (Saharia et al., 2022) trained on the ROCO dataset (Pelka et al., 2018) as a data augmentation strategy. In our initial experiments, we generate 1,000 images per gender and class for augmentation. The details of Chest X-ray dataset and the generation details are introduced in Appendix D.

However, despite using an equal quantity of generated images for each demographic group, fairness issues in the classification model not only persist but, as shown in Table 1, even worsen. This is evidenced by higher values of $\mathrm{Avg}(\Delta\mathrm{AUC})$ and $\max(\Delta\mathrm{AUC})$, calculated as

$$\mathrm{Avg}(\Delta\mathrm{AUC}) = \frac{1}{|\mathcal{C}|} \sum_{c \in \mathcal{C}} |\mathrm{AUC}_c^{\mathrm{male}} - \mathrm{AUC}_c^{\mathrm{female}}|, \quad \max(\Delta\mathrm{AUC}) = \max_{c \in \mathcal{C}} |\mathrm{AUC}_c^{\mathrm{male}} - \mathrm{AUC}_c^{\mathrm{female}}|,$$

where $\mathcal{C}$ denotes the set of classes. These results imply that generated images may exacerbate fairness issues, likely due to quality discrepancies across demographic groups.

*Table 1.* Comparison of classification performance and fairness metrics using different data augmentation strategies on the Chest X-ray dataset. **Blue** indicates an improvement in fairness, while **Red** denotes a deterioration compared to the baseline. All augmented data are generated by a text-to-medical-image model, with Fair and Unfair subsets selected from the entire generated dataset using Algorithm 1. Full augmentation worsens fairness, suggesting quality bias issues in the generated images. Data augmentation with the Fair Subset uses generated data of equal quality across genders, identified by lower DQA scores, yields lower $\mathrm{Avg}(\Delta\mathrm{AUC})$ and $\max(\Delta\mathrm{AUC})$ values without applying any fairness-specific technique. This outcome suggests that DQA effectively identifies reliable evaluation metrics for assessing fairness in generated image quality.

| | OVERALL AUC | AUC^MALE | AUC^FEMALE | AVG($\Delta$AUC) ↓ | max($\Delta$AUC) ↓ | $\overline{\mathrm{DQA}}$ |
|---|---|---|---|---|---|---|
| BASELINE | 53.33 | 55.30 | 50.58 | 6.30 | 16.80 | - |
| FULL AUGMENTATION | 54.39 | 56.55 | 51.39 | **6.76** | **16.64** | - |
| FAIR SUBSET (LOWER DQA) | 53.91 | 55.84 | 51.24 | **6.19** | **15.72** | 0.0868 |
| UNFAIR SUBSET (HIGHER DQA) | 54.32 | 56.40 | 51.43 | **6.71** | **17.19** | 0.5495 |

## A.2. Validity of DQA

To validate the effectiveness of DQA in identifying reliable image encoders for quality assessment, we construct both fair and unfair generated datasets in terms of quality as identified by their DQA scores. The fair generated dataset is expected to enhance fairness in classification when used for data augmentation, while the unfair generated dataset is anticipated to exacerbate fairness issues.

These datasets are characterized by lower (fair) and higher (unfair) DQA scores, evaluated using a reliable image encoder $f^*$. Specifically, let $A_{\mathrm{gen}}$ and $B_{\mathrm{gen}}$ represent two groups of generated data, with subsets $S_A \subset A_{\mathrm{gen}}$ and $S_B \subset B_{\mathrm{gen}}$, each of size $k = 0.2 \times |A_{\mathrm{gen}}|$. We define the fair and unfair subsets as $(S_A^{\mathrm{fair}}, S_B^{\mathrm{fair}}) = \arg\min_m \mathrm{DQA}(S_A^{(m)}, S_B^{(m)}; f^*)$ and $(S_A^{\mathrm{unfair}}, S_B^{\mathrm{unfair}}) = \arg\max_m \mathrm{DQA}(S_A^{(m)}, S_B^{(m)}; f^*)$, selected from $M$ candidate subsets $\{(S_A^{(m)}, S_B^{(m)})\}_{m=1}^M$.

To construct meaningful candidate pairs, we employ influence scores as a probabilistic measure of each image's impact on the DQA score, calculated via influence functions (Cook & Weisberg, 1980). These scores are normalized and used in

a multinomial sampling scheme, allowing us to prioritize high-impact images in both fair and unfair selection processes. Algorithm 1 in Appendix A.3 details the steps for sampling fair and unfair subsets, using influence-based probabilities to guide the selection.

For the classification task, we train a ResNet-50 model on the Chest X-ray diagnosis dataset, as outlined in Sec. A.1. Initial experiments in Sec. A.1 used an augmentation set containing 1000 images per gender and class. For DQA-guided augmentation, we add either the fair subset $(S_A^{\text{fair}}, S_B^{\text{fair}})$ or the unfair subset $(S_A^{\text{unfair}}, S_B^{\text{unfair}})$, each consisting of 200 images per gender and class, to assess how these augmentations impact model performance and demographic fairness. This setup enables a comparative evaluation of overall accuracy and fairness across demographic groups, thereby justifying the validity of DQA as an indicator of reliability.

The experimental results, shown in Table 1, demonstrate the effectiveness of the DQA score: the fair subset identified by low DQA improves fairness in classification AUC scores across demographic groups, even though DQA is not specifically designed for classification fairness, whereas the unfair subset (high DQA) worsens fairness outcomes.

### A.3. Fair/Unfair Subset Sampling Algorithm with DQA

---

**Algorithm 1** Finding Fair and Unfair Subsets Using Influence Scores for DQA

---

1: **Input:** Generated datasets $A_{\text{gen}}$ and $B_{\text{gen}}$; reference datasets $A_{\text{ref}}$ and $B_{\text{ref}}$; reliable encoder $f^*$; subset size $k$; number of samples $M$; small constant $\epsilon$

2: **Output:** Fair/Unfair subsets $(S_A^{\text{fair}}, S_B^{\text{fair}}), (S_A^{\text{unfair}}, S_B^{\text{unfair}})$

3: $F_A, F_B, F_{A_{\text{ref}}}, F_{B_{\text{ref}}} \leftarrow \{f^*(x_i) \mid x_i \in A_{\text{gen}}, B_{\text{gen}}, A_{\text{ref}}, B_{\text{ref}}\}$

4: $\text{DQA}_{\text{original}} \leftarrow \text{DQA}(F_A, F_B, F_{A_{\text{ref}}}, F_{B_{\text{ref}}})$

5: **for** each $x_i \in A_{\text{gen}}$ and $x_j \in B_{\text{gen}}$ **do**

6:     $F_A^{-i}, F_B^{-j} \leftarrow F_A \setminus \{f^*(x_i)\}, F_B \setminus \{f^*(x_j)\}$

7:     $\delta_i^A \leftarrow \text{DQA}_{\text{original}} - \text{DQA}(F_A^{-i}, F_B, F_{A_{\text{ref}}}, F_{B_{\text{ref}}})$

8:     $\delta_j^B \leftarrow \text{DQA}_{\text{original}} - \text{DQA}(F_A, F_B^{-j}, F_{A_{\text{ref}}}, F_{B_{\text{ref}}})$

9: **end for**

10: **Adjust influence scores for sampling:**

11: For fair subsets, invert influence scores:

12: $p_i^{A,\text{fair}}, p_j^{B,\text{fair}} \leftarrow \frac{-\delta_i^A - \min\{-\delta_i^A\} + \epsilon}{\sum_i (-\delta_i^A - \min\{-\delta_i^A\}) + \epsilon}, \frac{-\delta_j^B - \min\{-\delta_j^B\} + \epsilon}{\sum_j (-\delta_j^B - \min\{-\delta_j^B\}) + \epsilon}$

13: For unfair subsets, use original influence scores:

14: $p_i^{A,\text{unfair}}, p_j^{B,\text{unfair}} \leftarrow \frac{\delta_i^A - \min\{\delta_i^A\} + \epsilon}{\sum_i (\delta_i^A - \min\{\delta_i^A\}) + \epsilon}, \frac{\delta_j^B - \min\{\delta_j^B\} + \epsilon}{\sum_j (\delta_j^B - \min\{\delta_j^B\}) + \epsilon}$

15: **Initialize:** $\text{best\_DQA} \leftarrow \infty$, $\text{worst\_DQA} \leftarrow -\infty$

16: **for** $m = 1$ to $M$ **do**

17:     **Sample fair/unfair candidate subsets:**

18:     $S_A^{(m,\text{fair})}, S_B^{(m,\text{fair})} \leftarrow \text{Sample}(A_{\text{gen}}, k, p_i^{A,\text{fair}}), \text{Sample}(B_{\text{gen}}, k, p_j^{B,\text{fair}})$

19:     $\text{DQA}^{(m,\text{fair})} \leftarrow \text{DQA}(S_A^{(m,\text{fair})}, S_B^{(m,\text{fair})}, F_{A_{\text{ref}}}, F_{B_{\text{ref}}})$

20:     **Compute DQA for fair/unfair candidate:**

21:     **if** $\text{DQA}^{(m,\text{fair})} < \text{best\_DQA}$ **then**

22:         $\text{best\_DQA} \leftarrow \text{DQA}^{(m,\text{fair})}$

23:         $(S_A^{\text{fair}}, S_B^{\text{fair}}) \leftarrow (S_A^{(m,\text{fair})}, S_B^{(m,\text{fair})})$

24:     **end if**

25:     $S_A^{(m,\text{unfair})}, S_B^{(m,\text{unfair})} \leftarrow \text{Sample}(A_{\text{gen}}, k, p_i^{A,\text{unfair}}), \text{Sample}(B_{\text{gen}}, k, p_j^{B,\text{unfair}})$

26:     $\text{DQA}^{(m,\text{unfair})} \leftarrow \text{DQA}(S_A^{(m,\text{unfair})}, S_B^{(m,\text{unfair})}, F_{A_{\text{ref}}}, F_{B_{\text{ref}}})$

27:     **if** $\text{DQA}^{(m,\text{unfair})} > \text{worst\_DQA}$ **then**

28:         $\text{worst\_DQA} \leftarrow \text{DQA}^{(m,\text{unfair})}$

29:         $(S_A^{\text{unfair}}, S_B^{\text{unfair}}) \leftarrow (S_A^{(m,\text{unfair})}, S_B^{(m,\text{unfair})})$

30:     **end if**

31: **end for**

32: **Return:** $(S_A^{\text{fair}}, S_B^{\text{fair}}), (S_A^{\text{unfair}}, S_B^{\text{unfair}})$

---

## B. Details of Synthetic Data in Figure 3

To construct the synthetic dataset, we generated non-Gaussian data for groups $A$ and $B$ by combining multivariate normal and exponential distributions. Each group has distinct means, covariances, and exponential scaling factors to ensure variability and non-Gaussian characteristics in the data. For group $A$, we define the mean as $\boldsymbol{\mu}_A$ and covariance as $\Sigma_A$. Samples for group $A$ were drawn from a multivariate normal distribution, $\mathcal{N}(\boldsymbol{\mu}_A, \Sigma_A)$, and combined with exponential noise with a scale parameter $\lambda_A$. Similarly, for group $B$, we define the mean as $\boldsymbol{\mu}_B$ and covariance as $\Sigma_B$. Samples are drawn from $\mathcal{N}(\boldsymbol{\mu}_B, \Sigma_B)$ and combined with exponential noise with a scale parameter $\lambda_B$.

$$A_{\text{ref}} = \mathcal{N}(\boldsymbol{\mu}_A, \Sigma_A) + \text{Exp}(\lambda_A)$$
$$B_{\text{ref}} = \mathcal{N}(\boldsymbol{\mu}_B, \Sigma_B) + \text{Exp}(\lambda_B)$$

To introduce distribution shift as examples for fair and unfair case, translations are applied to each group. Let $\mathbf{t}_A$ and $\mathbf{t}_B$ represent the translations for groups $A$ and $B$ respectively. The test data for each group is generated as:

$$A_{\text{gen}} = \mathcal{N}(\boldsymbol{\mu}_A, \Sigma_A) + \mathbf{t}_A + \text{Exp}(\lambda_A)$$
$$B_{\text{gen}} = \mathcal{N}(\boldsymbol{\mu}_B, \Sigma_B) + \mathbf{t}_B + \text{Exp}(\lambda_B)$$

where $\boldsymbol{\mu}_A = [\mu_{A1}, \mu_{A2}]$ and $\Sigma_A = \begin{bmatrix} \sigma_{A1}^2 & 0 \\ 0 & \sigma_{A2}^2 \end{bmatrix}$ denote the mean and covariance of group $A$, $\boldsymbol{\mu}_B = [\mu_{B1}, \mu_{B2}]$ and $\Sigma_B = \begin{bmatrix} \sigma_{B1}^2 & 0 \\ 0 & \sigma_{B2}^2 \end{bmatrix}$ denote the mean and covariance of group $B$, $\lambda_A$ and $\lambda_B$ represent the exponential scaling factors for groups $A$ and $B$, and $\mathbf{t}_A$ and $\mathbf{t}_B$ are translations applied to groups $A$ and $B$, respectively.

Using this structure, we introduce non-Gaussianity through the combination of multivariate normal and exponential distributions with group-specific parameters $\boldsymbol{\mu}_A, \Sigma_A, \lambda_A$, and $\boldsymbol{\mu}_B, \Sigma_B, \lambda_B$. Test (generated) datasets maintain only the mean parameters for each group, but covariance and scaling factors are shifted as well as translations to mimic the distribution shift in generative models.

For the reference set, we choose $\mu_{A1} = \mu_{A2} = 0$, $\sigma_{A1}^2 = \sigma_{A2}^2 = 1$, $\lambda_A = 1$, $\mu_{B1} = \mu_{B2} = 15$, $\sigma_{B1}^2 = \sigma_{B2}^2 = 8$, and $\lambda_B = 2$. For the generated set, we change the covariance as $\sigma_{A1}^2 = \sigma_{A2}^2 = 3$ and $\sigma_{B1}^2 = \sigma_{B2}^2 = 12$, and shift the scaling $\lambda_A \leftarrow \lambda_A + 0.2$, and $\lambda_B \leftarrow \lambda_B + 0.2$. Moreover, we apply different scaling and translations for fair and unfair synthetic dataset. Specifically, we choose $\mathbf{t}_A = [3, 3]$ and $\mathbf{t}_B = [-3, -3]$, to depict a fair scenario, while $\mathbf{t}_A = [1, 1]$ and $\mathbf{t}_B = [-11, -11]$ are chosen to simulate unfairly skewed distribution for group $B$.

## C. Constructing Evaluation Dataset for DQA

We consider realistic scenarios encountered in text-to-image generation for human image datasets using Stable Diffusion Inpainting (Rombach et al., 2022). Our baseline follows the recommended settings from (Lui et al., 2024), where image quality degradation is achieved by adjusting specific hyperparameters. Specifically, the baseline parameters include a sampling step size of $T = 40$, noise strength $s_n = 0.7$, guidance scale $s_g = 7.5$, and a refinement phase during the last 20% of sampling, denoted by $\tau_{\text{refine}} = 0.2$. The scenarios we evaluate are as follows:

1. **Baseline**: Uses sufficient diffusion steps with a balanced influence between the initial image and noise, with parameters $(T, s_n, s_g, \tau_{\text{refine}}) = (40, 0.7, 7.5, 0.2)$.

2. **Weak Guidance**: Reduces the guidance scale, weakening the model's adherence to the text prompt. This can result in images that lack coherence or do not fully align with the desired content, $(T, s_n, s_g, \tau_{\text{refine}}) = (40, 0.7, \mathbf{1.0}, 0.2)$.

3. **Fewer Steps**: Halves the number of diffusion steps compared to the baseline, reducing the model's capacity to refine details and potentially resulting in noisier outputs, $(T, s_n, s_g, \tau_{\text{refine}}) = (\mathbf{20}, 0.7, 7.5, 0.2)$.

4. **Strong Noise**: Increases the noise strength, introducing more randomness and potentially causing the image to deviate from the prompt, $(T, s_n, s_g, \tau_{\text{refine}}) = (40, \mathbf{0.9}, 7.5, 0.2)$.

5. **No Refiner**: Omits the refinement phase, leading to images with fewer details and a less polished appearance, $(T, s_n, s_g, \tau_{\text{refine}}) = (40, 0.7, 7.5, \mathbf{0.0})$.

6. **Combination**: Combines weak guidance, fewer steps, and strong noise, creating highly degraded images, $(T, s_n, s_g, \tau_{\text{refine}}) = (\mathbf{20}, \mathbf{0.9}, \mathbf{1.0}, \mathbf{0.0})$.

We select 10 professions commonly referenced in the literature (Lui et al., 2024; Gustafson et al., 2023; Cho et al., 2023), including flight attendant, nurse, secretary, teacher, veterinarian, engineer, pilot, firefighter, surgeon, and builder. Additionally, we considered four racial groups identified in (Lui et al., 2024): Asian, Black, Indian, and White Caucasian. The examples of constructed datasets are visualized in Figure 12 in the last page.

## D. Details in Chest X-ray Dataset and Generation

### D.1. Details of the Chest X-ray Dataset

We use the NIH ChestX-ray14 dataset (Wang et al., 2017), a large repository containing 112,120 chest X-ray images from 30,805 patients, annotated with 14 common thoracic disease categories, including Hernia, Pneumonia, Fibrosis, Emphysema, Edema, Cardiomegaly, Pleural Thickening, Consolidation, Mass, Pneumothorax, Nodule, Atelectasis, Effusion, and Infiltration. By including 'No Findings' as a benign case, the dataset expands to 15 classes. It also includes demographic information, with approximately 56.5% male and 43.5% female patients.

### D.2. Details of Synthetic Chest X-ray Generation

To generate synthetic Chest X-ray images, we use a pre-trained ImageGen model (Saharia et al., 2022) trained on the ROCO dataset (Pelka et al., 2018), which contains paired image and text data for medical purposes. The pretrained model is available on HuggingFace (Wolf, 2019) under the model ID Nihirc/Prompt2MedImage. We generate 1,000 images per gender and class, resulting in a total of 30,000 images across 2 genders and 15 classes. The input prompt format for generation is "Chest X-ray image of a {GENDER} patient showing a/an {DISEASE}."

## E. DQA analysis for Medical Image

### E.1. Constructing Reference Dataset for Medical Image

In the medical image, we utilize the Chest X-ray diagnosis dataset in Sec. A.1 as the reference, given its consistent image quality across genders, controlled through human annotations. This consistency makes it an effective benchmark for quality assessment. Specifically, we designate the training set of Chest X-ray images as the reference dataset, while the test set and its transformations are used as a mimic of the generated dataset to help identify a reliable image encoder. In more detail, the real test data remains in-distribution relative to the training dataset, while we simulate generative model failures (Borji, 2023) by applying transformations to the test set, creating poor-quality images as shown in Fig. 8 (a).

### E.2. Reliability Analysis for Image Encoders for Medical Image

For medical images, we assess encoders such as InceptionV3 and RN50 pretrained on IN-1K, alongside RN50 models trained directly on the Chest X-ray dataset using supervised learning, self-supervised learning (SimCLR) (Chen et al., 2020), and supervised learning on a single-gender subset. The RN50 pretrained on IN-1K achieves the lowest DQA score, suggesting that pretraining on a diverse dataset helps mitigate biases inherent in domain-specific data. In contrast, models trained directly on medical images exhibit higher DQA scores, potentially due to the amplification of existing biases within the specialized dataset.

## F. DQA-Guidance for Medical Image

### F.1. Experimental Details

To verify the effectiveness of DQA-Guidance in mitigating quality bias, we utilize a medical dataset and a generative model for medical images, consistent with the setup in previous sections. Specifically, we apply Eq. (4) to the text-to-medical-image model during the sampling stage, generating 100 images per gender and class, resulting in a total of 3000 images (2 genders and 15 classes). For each gender, the prompt "Chest X-ray image of a {GENDER} patient showing a {DISEASE_NAME}." is used, with the Chest X-ray training data for each gender serving as a reference to compute empirical DQA during the sampling stage. In the experiments, we vary $\lambda_1$ while fixing $\lambda_2 = 0$ to examine the impact of

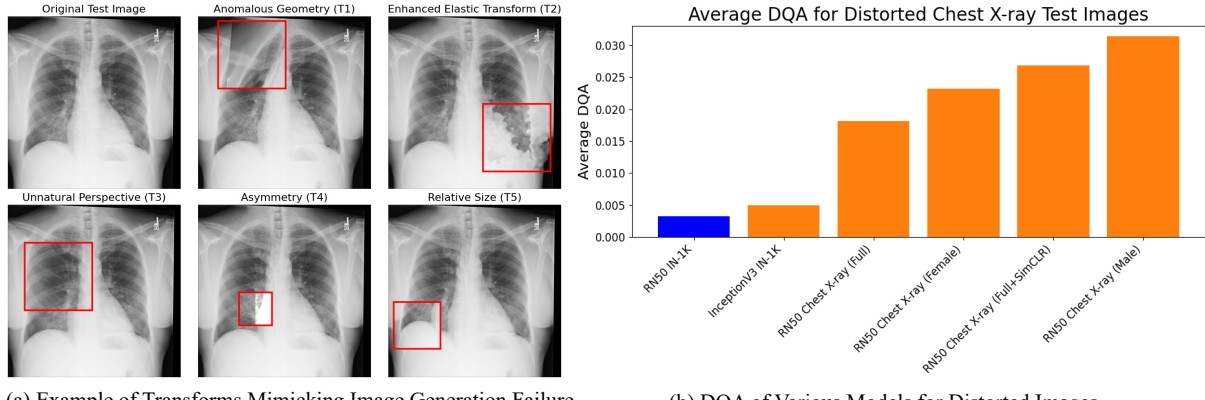

(a) Example of Transforms Mimicking Image Generation Failure

(b) DQA of Various Models for Distorted Images

*Figure 8.* (a) To assess the DQA across varying qualities of generated medical images, we simulate generative model failures by applying transformations to test images that reflect common failure patterns in generative models. (b) By incrementally applying these transformations and evaluating the reliability of various pretrained encoders, we find that a ResNet-50 model pretrained on ImageNet-1K demonstrates greater reliability in quality assessment, consistently handling poor-quality images across demographic groups by showing lowest DQA in average. In contrast, the same model trained on reference data shows higher DQA scores, indicating unreliable image quality assessment.

DQA-Guidance on both generation quality and the quality gap between groups.

### F.2. Result Analysis for DQA-Guidance

Fig. 9 demonstrates the clear impact of DQA-Guidance on medical image generation. Compared to the baseline ($\lambda_1 = 0$), increasing $\lambda_1$ effectively reduces quality disparities in generated images while substantially improving overall image quality. However, setting $\lambda_1$ too high introduces excessive noise, leading to a decline in image quality. These findings suggest that DQA not only provides a reliable measure for evaluating fairness but also serves as an effective regularizer, enhancing fairness in image generation when applied as guidance in diffusion models. Additionally, larger values of $\lambda_2$ intuitively contribute to improved generation quality. Qualitative results of DQA-Guidance is shown in Fig. 10. Similar to DQA-Guidance for human images, the improvements primarily focus on refining texture. While these improvements may appear subtle from a user's perspective, the measured quality confirms that the hyperparameters $\lambda_1$ and $\lambda_2$ play a significant role in enhancing overall quality and reducing quality disparities.

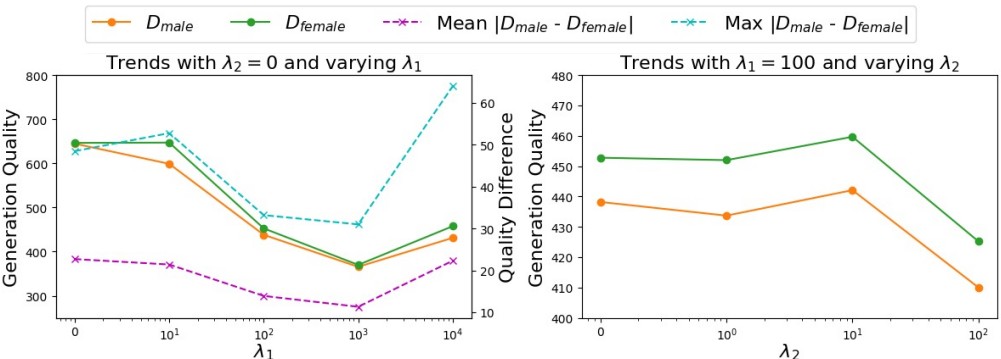

*Figure 9.* Experimental results for generation quality and quality disparities with DQA-Guidance. The left plot shows the impact of $\lambda_1$ on generation quality for each demographic group in Chest X-ray image generation (lower values indicate better quality) and displays the average and maximum quality gap across all disease classes (lower values indicate reduced disparity). The right plot illustrates the effect of $\lambda_2$ on overall generation quality. Here, $\lambda_1 = 0$ denotes no DQA guidance, while higher $\lambda_1$ values reflect a stronger influence of DQA-Guidance. DQA-Guidance effectively enhances generation quality and reduces quality disparities across demographic groups.

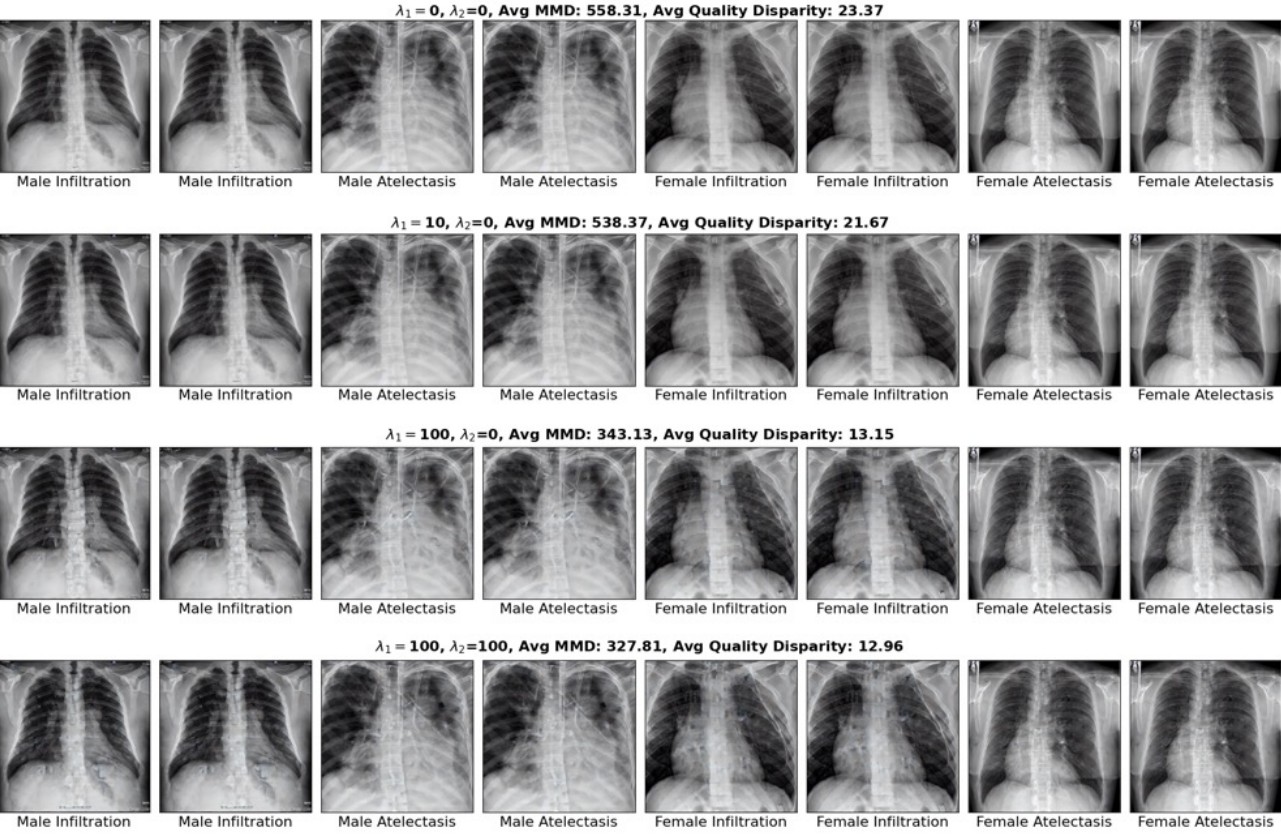

*Figure 10.* Qualitative results of DQA-Guidance for medical image generation. The examples highlight improvements primarily in texture refinement, demonstrating the method's ability to enhance overall image quality while addressing disparities across different conditions.

## G. DQA on Different Types of Image Quality Assessment

In addition to our approach, other methods for assessing image quality include visual question answering (VQA) (Lui et al., 2024) and neural networks specifically trained for quality evaluation (Kolchinski et al., 2019; Tian et al., 2022; Chen et al., 2024a).

In (Lui et al., 2024), VQA models are asked questions such as Prompt 1: "Is this image real or fake?" or Prompt 2: "Are this person's limbs distorted?" to detect unreal aspects of a given image. However, as the image encoder used in VQA models may exhibit bias, the distribution of VQA answers could also be biased. To quantify this bias, we adapt DQA in Eq. (1) by replacing $D(f(\cdot), f(\cdot))$ with $p(h(\cdot), \mathcal{T})$, where $h$ denotes the VQA model and $p$ represents the probability of detecting abnormalities based on the text prompt $\mathcal{T}$. This approach utilizes the probability of realism detected by the VQA model as the image quality assessment metric.

$$\text{DQA}^{\text{VQA}} = \frac{|p(h(A_{\text{gen}})) - p(h(B_{\text{gen}}))|}{p(h(\mathcal{I}_{\text{gen}}))}$$

We also adapt DQA to image quality assessment (IQA) models that output indicators of general image quality. For example, TOPIQ (Chen et al., 2024a) is a supervised network designed for image quality evaluation. It is trained on datasets such as FLIVE (Ying et al., 2020) for general images or CGFIQA (Chen et al., 2024b) for facial images, using a regression task to predict quality scores. Let $s(\cdot)$ an IQA model's outcome, then we adapt DQA in Eq. (1) by replacing $D(f(\cdot), f(\cdot))$ with $\bar{s}(\cdot)$, the mean of quality score over each group.

$$\text{DQA}^{\text{IQA}} = \frac{|\bar{s}(A_{\text{gen}}) - \bar{s}(B_{\text{gen}})|}{\bar{s}(\mathcal{I}_{\text{gen}})}$$

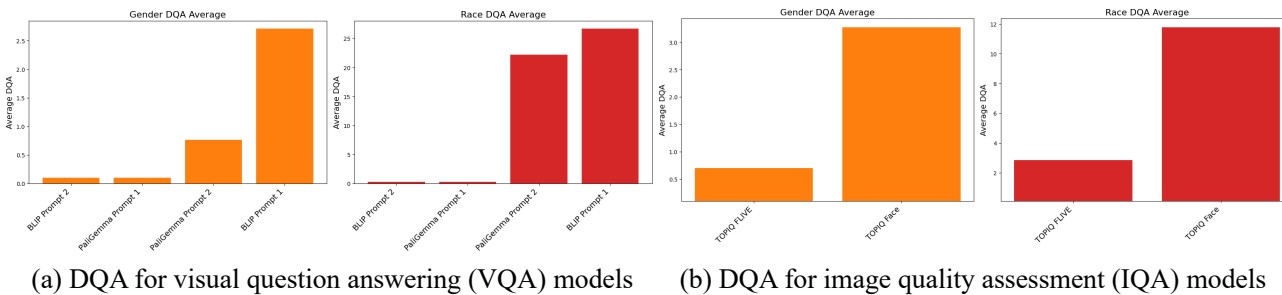

(a) DQA for visual question answering (VQA) models     (b) DQA for image quality assessment (IQA) models

*Figure 11.* DQA on different types of image quality assessments. We compare DQA scores for gender and racial fairness across VQA models (BLIP and PaliGemma) under two prompts, as well as IQA models trained on general and facial datasets. Results highlight varying tendencies in DQA across models and prompts, with racial fairness remaining a significant challenge and facial dataset-trained IQA models showing higher DQA scores.

To summarize the quality assessment methods utilized throughout the paper:

- **Distance-based methods**: Measure the similarity between the feature distributions of generated images and real images to determine image quality (e.g., FID).

- **VQA-based methods**: Assess visual realism and detect whether images are free from noticeable distortions or errors.

- **General IQA methods**: Evaluate objective image quality metrics such as blur, noise, sharpness, and color saturation.

We use BLIP (Li et al., 2022) and PaliGemma (Beyer et al., 2024) as representative VQA models with two different prompts. Additionally, we utilize two pre-trained versions of TOPIQ for general IQA: one trained on the FLIVE dataset for general images and another trained on the CGFIQA dataset for facial images.

The experimental results for these different types of image quality assessments are visualized in Fig. 11. Interestingly, VQA models exhibit varying tendencies. For gender-based DQA, PaliGemma demonstrates reliability with low DQA for Prompt 1 but shows relatively high DQA for Prompt 2. Conversely, BLIP achieves reliable results with Prompt 2 but exhibits high DQA for Prompt 1. For racial DQA, both models exhibit similar tendencies with gender-based DQA; however, the overall DQA values are significantly higher, indicating that racial bias remains a pressing concern in fair evaluation.

In the case of IQA models, the version trained on a general dataset exhibits greater reliability with low DQA, whereas the version trained on facial datasets demonstrates significantly higher DQA. This result highlights potential challenges in achieving fairness when applying models trained on specific datasets.

# H. Impact of DQA-Guidance on Downstream Tasks

In line with Sec. A.2, we further investigate the impact of DQA-Guidance on fairness in AUC across gender in medical image classification. We compare the classification performance using different versions of generated samples. For this analysis, we use 100 images per gender and class as augmentation, while Table 1 reports results based on 1,000 images per gender and class for full augmentation and 200 images per gender and class for fair and unfair subsets.

Table 2 shows the classification performance when generative samples created with DQA-Guidance are used for data augmentation. To isolate the impact of $\lambda_1$, we eliminate the influence of $\lambda_2$ by setting $\lambda_2 = 0$.

Compared to baseline augmentation (No Guidance), DQA-Guidance improves the overall AUC and significantly reduces both the mean and maximum AUC gaps between demographic groups. This enhancement is achieved without explicit fairness constraints, relying solely on improved quality parity between groups.

*Table 2.* Classification performance and fairness metrics on the Chest X-ray dataset using DQA-Guidance for data augmentation. The table compares results across augmentation strategies using 100 images per gender and class. $\lambda_1$ is varied while $\lambda_2$ is set to 0 to isolate its effect. Compared to No Guidance, DQA-Guidance improves overall AUC and significantly reduces both the mean and maximum AUC gaps between demographic groups, demonstrating its effectiveness in enhancing quality parity without applying explicit fairness constraints.

|  | OVERALL AUC | AUC$^{\text{MALE}}$ | AUC$^{\text{FEMALE}}$ | AVG($\Delta$AUC) $\downarrow$ | max($\Delta$AUC) $\downarrow$ |
|---|---|---|---|---|---|
| BASELINE (NO AUGMENTATION) | 53.33 | 55.30 | 50.58 | 6.30 | 16.80 |
| NO GUIDANCE | 54.22 | 56.48 | 51.08 | 6.90 | 16.87 |
| DQA-GUIDANCE $\lambda_1 = 10$ | 54.31 | 56.37 | 51.45 | 6.55 | 16.31 |
| DQA-GUIDANCE $\lambda_1 = 20$ | 54.31 | 56.19 | 51.69 | 6.46 | 16.30 |
| DQA-GUIDANCE $\lambda_1 = 100$ | 54.37 | 56.36 | 51.60 | 6.56 | 16.27 |

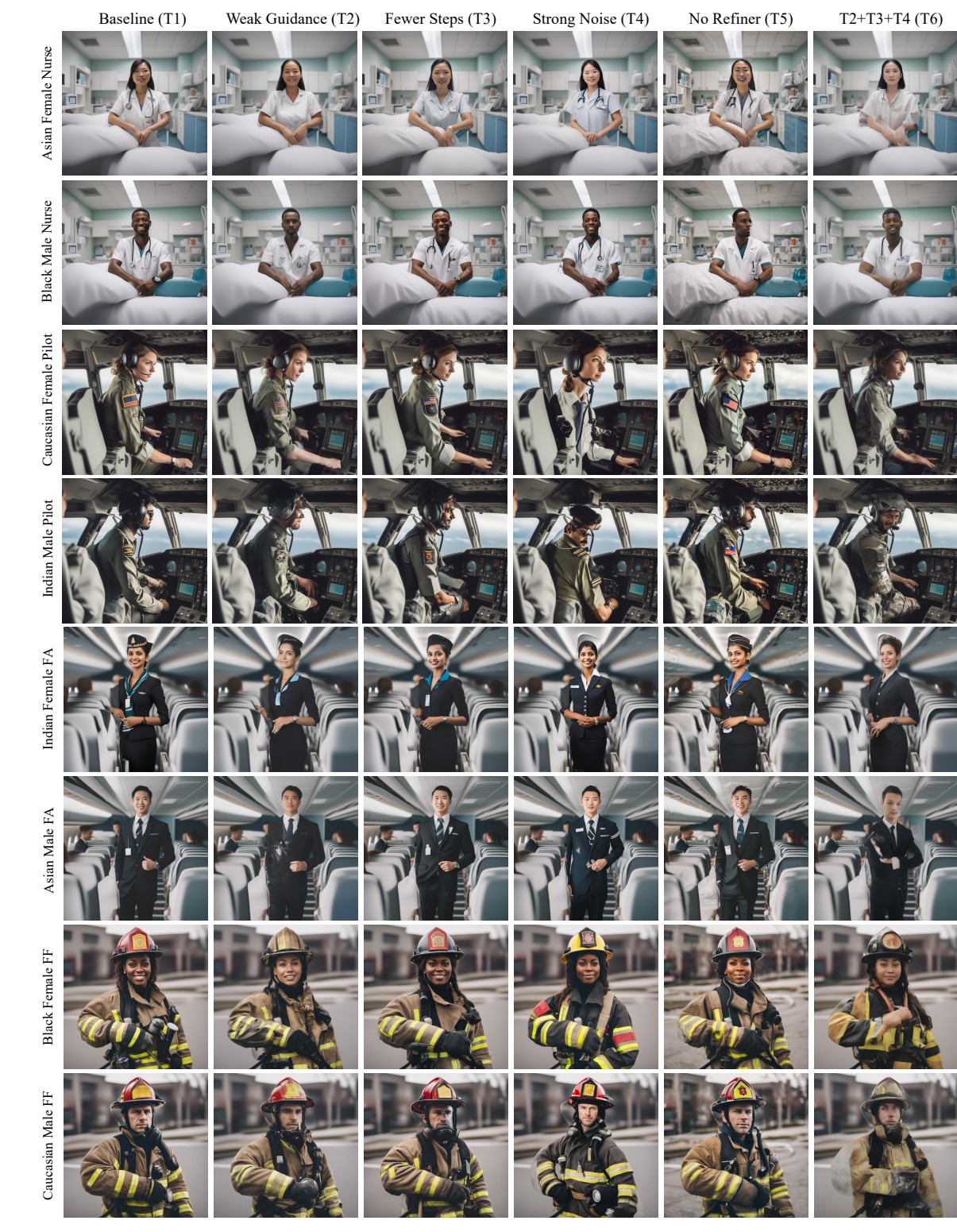

*Figure 12.* Examples of constructed evaluation datasets for DQA under various text-to-image generation scenarios to controlled degradation of generated image. The scenarios include Baseline, Weak Guidance (T2), Fewer Steps (T3), Strong Noise (T4), No Refiner, and a combination of T2, T3, and T4. Each setting adjusts specific hyperparameters of Stable Diffusion Inpainting (Rombach et al., 2022) to simulate realistic degradations in image quality. The datasets represent 10 professions and 4 racial groups, illustrating the diversity and quality variations used for evaluation while four professions (Nurse, Pilot, Flight Attendant (FA), and fire fighter (FF)) are presented in the example.

