# OpenReview forum: "Reliable Image Quality Evaluation and Mitigation of Quality Bias in Generative Models"
_ICML.cc/2025/Conference — Submitted to ICML 2025_

### Official Review · Reviewer_KtiY · 2025-03-09

**Overall Recommendation:** 3

**Summary:**

This paper introduces the Difference in Quality Assessment (DQA) score, which is designed to evaluate the reliability of evaluation metrics such as the Fréchet Inception Distance (FID). Additionally, the DQA framework aids in identifying more reliable image encoders, thereby enhancing the robustness of evaluation metrics. Furthermore, the proposed DQA-Guidance method not only improves the quality of pretrained diffusion models but also advances fairness.

## update after rebuttal
Overall, the paper is well-written. The method is novel and the results are good. However, I don’t think it quite reaches the level for an "Accept" (score 4). I keep my original score.

**Claims And Evidence:**

Yes

**Essential References Not Discussed:**

Regarding energy guidance, the paper 'Egsde: Unpaired Image-to-Image Translation via Energy-Guided Stochastic Differential Equations' should be cited, as it introduces the use of energy guidance to improve generative models.

**Experimental Designs Or Analyses:**

Yes, I have carefully reviewed the experimental section and all the figures and tables in the main body of the paper. I believe the authors’ experimental design is sound and addresses my concerns step by step. However, I have one question: Can the DQA framework be used to evaluate the performance of currently popular text-to-image diffusion models, such as Flux?

**Methods And Evaluation Criteria:**

Yes

**Other Comments Or Suggestions:**

No.

**Other Strengths And Weaknesses:**

Strengths:
- The FID (Fréchet Inception Distance) is a widely used metric for evaluating generative models. However, this paper highlights some unreliable phenomena associated with FID during the evaluation process, particularly its mismatch with generation quality and fairness. Notably, better FID scores may sometimes correspond to worse generation quality.
- The introduction of the DQA (Difference in Quality Assessment) method is a significant contribution, as it helps identify more reliable image encoders. This is crucial for the appropriate selection of evaluation metrics.
- The proposal of DQA-Guidance is another key strength, as it enhances generative models by improving generation quality while maintaining fairness.

Weakness:
see question seciton.

**Questions For Authors:**

1. In the DGA-Guidance section, regarding Equation (3), could you provide further clarification on how Group A and Group B are determined?
2. Can the DQA framework be used to evaluate the performance of currently popular text-to-image diffusion models, such as Flux?

**Relation To Broader Scientific Literature:**

The main contributions of this paper can be broadly summarized as the introduction of the Difference in Quality Assessment (DQA) score to evaluate the reliability of existing evaluation metrics, followed by the application of the DQA score to enhance diffusion models. To my knowledge, other related works tend to focus more on constructing comprehensive benchmarks for evaluating generative models. In this regard, the approach presented in this paper appears to be novel.

**Theoretical Claims:**

No, I have not checked the theoretical proofs.

---

> ### Author Rebuttal · Authors · 2025-04-01
>
> ## Additional Reference
> Thanks for suggesting a missing reference. Although our paper already includes references related to energy-based guidance in text-to-image models—such as Composing Diffusion Models [1], Self-Guidance [2], and Universal Guidance [3]—the suggested reference [4] is indeed a valuable cornerstone in the energy-based guidance literature. We will include this reference in the revised version of our paper.
>
>
> [1] Liu, N., Li, S., Du, Y., Torralba, A., & Tenenbaum, J. B. (2022, October). Compositional visual generation with composable diffusion models. In European Conference on Computer Vision (pp. 423-439). Cham: Springer Nature Switzerland.
> [2] Epstein, D., Jabri, A., Poole, B., Efros, A., & Holynski, A. (2023). Diffusion self-guidance for controllable image generation. Advances in Neural Information Processing Systems, 36, 16222-16239.
> [3] Bansal, A., Chu, H. M., Schwarzschild, A., Sengupta, S., Goldblum, M., Geiping, J., & Goldstein, T. (2023). Universal guidance for diffusion models. In Proceedings of the IEEE/CVF Conference on Computer Vision and Pattern Recognition (pp. 843-852).
> [4] Zhao, M., Bao, F., Li, C., & Zhu, J. (2022). Egsde: Unpaired image-to-image translation via energy-guided stochastic differential equations. Advances in Neural Information Processing Systems, 35, 3609-3623.
>
> ## Clarification on Equation 3
> Thank you for pointing out the lack of explicit notation in Equation 3. While Groups A and B are introduced as two demographic groups in Section 3.1, the page distance between that section and Equation 3 may cause confusion. To improve clarity, we agree that the definitions of Groups A and B should be stated explicitly near Equation 3.
>
> Specifically, Group A and Group B refer to demographic groups such as *male* and *female*. The terms $z_t^A$ and $z_t^B$ represent latent variables derived from the input prompt:
> *“a photo of a {GENDER} who works as a {PROFESSION}.”*
>
> Moreover, Groups A and B can also represent other demographic attributes, such as different races, as demonstrated in our rebuttal to Reviewer h2hy. We expect that this formulation can be extended further to accommodate any desired quality bias to mitigate, depending on the fairness objective.
>
> ## Extension to Other Generative Models
>
> Yes, the DQA framework can be applied to any generative model such as Flux. Since DQA serves as a reliability measure for evaluation metrics such as FID, it does not rely on the performance of the generative model itself—as long as the model produces images and their quality is evaluated using FID.
>
> Moreover, the DQA-Guidance approach is also model-agnostic and can be applied to any diffusion-based generative models.

---

> > ### Comment · Reviewer_KtiY · 2025-04-02
> >
> > Thank you for your response. I maintain my original score.

---

### Official Review · Reviewer_h2hy · 2025-03-14

**Overall Recommendation:** 4

**Summary:**

This paper proposes a Difference in Quality Assessment (DQA) measure that quantifies the reliability of existing quality evaluation metrics for generative models. The authors present a problem in generation model evaluation, i.e., the demographic bias. They find that conventional quality assessment measures are biased across groups, and the reasons can lie in the inappropriate reference selection or the inherent bias in the FID image encoder. They further apply the proposed DQA for guiding diffusion models to reduce cross-group quality discrepancies.

**Claims And Evidence:**

The claims in this paper are well-supported by literature and analyses, making them reasonable.

**Essential References Not Discussed:**

NA

**Experimental Designs Or Analyses:**

In section 5.3, the authors provide the performance evaluation of DQA-guidance on two sub-categories, i.e., male and female. Can you further provide more extensive results on other demographic groups?

**Methods And Evaluation Criteria:**

The proposed DQA and DQA-guidance are applicable to the research questions.

**Other Comments Or Suggestions:**

NA

**Other Strengths And Weaknesses:**

This paper provides comprehensive experimental results to support their findings and solutions. The article is well-written and easy to follow. A very good paper.

**Questions For Authors:**

See above.

**Relation To Broader Scientific Literature:**

NA

**Theoretical Claims:**

NA

---

> ### Author Rebuttal · Authors · 2025-04-01
>
> ## Extension of Demographic Groups
> Thank you for raising this point. Quality bias is not limited to gender; it also extends to other demographic attributes such as race. In our study, we consider four racial groups: Asian, Black, Caucasian, and Indian. We explore two possible directions for extending DQA-Guidance to handle multi-racial bias:
>
> 1. **Pairwise Group Comparison**
>    In this approach, we select two racial groups (e.g., Asian vs. Black) as Group A and Group B and apply DQA-Guidance in the same manner as the gender-bias case. This enables a detailed, pairwise analysis of racial bias and its mitigation.
>
> 2. **All-at-Once Comparison**
>    Alternatively, we can modify the DQA-Guidance formulation to consider all racial groups simultaneously by replacing the DQA term in Equation 4 with the average pairwise DQA across all race pairs, as defined below:
>    \\[
>    \tilde{\epsilon}_{\theta} (z_t) = \epsilon\_{\theta} (z_t) + \sigma\_t \nabla\_{z_t} \left(\lambda\_1 \, \text{AvgDQA} + \lambda\_2 \, D\big(f^*(\mathcal{I}\_{\text{gen}}), f^*(\mathcal{I}\_{\text{ref}})\big)\right)
>    \\]
>  where
>    \\[
>    \text{AvgDQA}(\mathcal{G}) = \frac{1}{\binom{n}{2}} \sum\_{1 \leq i < j \leq n} \text{DQA}(G\_i, G\_j; f^*)
>    \\]
>
>    This formulation generalizes fairness evaluation across all $n$ racial groups by averaging over all possible group pairs.
>
> For ease of implementation and to enable a more detailed analysis of individual racial quality disparities, we adopt the first approach in our additional experiments.
>
> |Stable Diffusion| Avg. MMD | Avg. MMD Disparity | Max MMD  Disparity | Worst Case                        |
> |----------------|----------|----------------|----------------|------------------------------------|
> | **Baseline**     | 118.36   | 14.49          | 38.14          | Caucasian vs Indian, Nurse         |
> | **DQA-Guidance** | 96.68    | 10.16          | 25.38          | Caucasian vs Indian, Nurse         |
>
> In our experiments, DQA-Guidance improves both the overall image quality and reduces quality disparity across racial groups, consistent with the results reported for the gender case.

---

### Official Review · Reviewer_yKK1 · 2025-03-14

**Overall Recommendation:** 2

**Summary:**

The paper aims to address the issue of quality disparities in image generation models, proposing the DQA score as a method for assessing the reliability of evaluation metrics, and introducing DQA-Guidance to mitigate quality bias in diffusion models. The core contributions are the DQA metric and its application to identify reliable image encoders and guide the diffusion sampling process.

## update after rebuttal ##
Thank you for the author's response. The limited scope of the comparison experiments remains my major concern. Therefore, I will maintain my original score.

**Claims And Evidence:**

- The idea of fairness in generative models and the biases of evaluation metrics have been discussed in previous works. The paper does not make a big step beyond the existing literature.

**Essential References Not Discussed:**

N/A

**Experimental Designs Or Analyses:**

- The experimental results for DQA-Guidance (Section 5.3 and Figure 6) are not compelling. The plots show only marginal improvements in image quality and quality disparity. The qualitative results in Figure 7 are subjective and do not provide strong evidence of the effectiveness of DQA-Guidance. The paper needs to provide a more comprehensive and objective evaluation of DQA-Guidance, including quantitative metrics beyond MMD.

- The paper does not adequately compare DQA and DQA-Guidance to existing fairness-aware evaluation metrics and mitigation techniques. It's unclear whether the proposed approach offers any significant advantages over existing methods.

**Methods And Evaluation Criteria:**

- The method for creating controlled datasets with varying degrees of image quality (Section 4.2 and Appendix C) is not sufficiently detailed. The specific hyperparameter adjustments and their impact on perceived image quality need to be better explained and justified. It's unclear if these adjustments consistently produce the intended quality gradations across different demographic groups.

**Other Comments Or Suggestions:**

- In several places, the wording is awkward or unclear. For example, the sentence "DQA serves not only as a reliability indicator for the evaluation metric but can also act as an energy function in generative models to regularize equal image quality across demographic groups" could be rephrased for clarity.

**Other Strengths And Weaknesses:**

Strengths：
+ The paper identifies a significant and underexplored problem of quality bias in generative models, which can have important practical implications.
+ The idea of using a controlled dataset to assess the reliability of evaluation metrics is promising, as it can uncover biases in the encoders.


Weaknesses
- The paper should include a more comprehensive set of evaluation metrics, including both quantitative and qualitative measures.
- The paper should compare the performance of DQA-Guidance to existing fairness-aware methods.

**Questions For Authors:**

See above

**Relation To Broader Scientific Literature:**

The DQA score attempts to formalize this by quantifying the discrepancy in quality assessments across demographic groups. However, the novelty lies in the specific formulation of DQA, not in the general concept of biased metrics. The paper needs to clearly articulate how DQA differs from and improves upon existing methods for detecting biases in evaluation, even if those methods are not directly applied to image quality. Is it more sensitive? More robust? Easier to interpret?

**Theoretical Claims:**

- The DQA score, as defined in Equation (1), lacks strong theoretical justification. The normalization by the denominator D(f(Igen), f(Iref)) is not adequately explained. It's unclear why this specific normalization is appropriate or how it ensures a reliable measure of bias. The paper needs to provide a more rigorous mathematical justification for the DQA formulation.

---

> ### Author Rebuttal · Authors · 2025-04-01
>
> ## Novelty of the Paper
> To the best of our knowledge, this paper is the first to address fairness issues in the evaluation metrics used for generated images.
>
> We distinguish two types of bias:
> - **(a) Bias in the evaluation metric**
> - **(b) Bias in quality in the generated image**
>
> Although (b) has been studied, (a) has not received adequate attention. In this work, we identify bias in a widely used evaluation metric, FID, and propose **DQA** as a reliability measure to detect and quantify such bias. By isolating evaluation bias through DQA by choosing a reliable image encoder, we can more accurately reveal the quality bias in generative models, (b). Furthermore, we introduce **DQA-Guidance**, a mitigation strategy that guides generative models toward reducing quality bias in the generated images.
>
> Taken together, our contributions are the first to highlight and address the two-fold fairness challenge in generative modeling.
> ## Regarding Comparison Method
> As ours is the first work to identify bias in evaluation metrics and to propose a framework for mitigating quality bias in generative models, we were unable to include direct comparisons with existing methods.
> ## Adjustment in the Controlled Dataset
> Please see the rebuttal for Reviewer 4STN.
> ## Theoretical Analysis for Equation 1
> The denominator captures the total generation shift, i.e., how far the generated distribution is from the reference distribution across all groups. It serves two purposes.
> 1. DQA measures how large the inter-group disparity is relative to the overall deviation. If both group-specific shifts are small, then even a small difference between them may be meaningful. Conversely, if the model globally generates low-quality outputs, a larger group disparity might be expected and less concerning.
> 2. Different generative models, encoders, and data domains may exhibit widely varying absolute distances. Without normalization, a group-level bias in numerator is hard to determined to be negligible or severe. Therefore, the denominator anchors the numerator to this global scale.
> ## Improvement via DQA-Guidance
> While Figures 6 and 7 illustrate the improvements in both performance and fairness of image quality, we add a table below for better clarity. This table explicitly demonstrates that an appropriate choice of $\lambda_1$ and $\lambda_2$ leads to substantial improvements in both fairness and overall image quality.
> ||Avg. MMD|Avg. MMD Diff.|Max MMD Diff.|
> |-|-|-|-|
> |Baseline|109.93|12.57|17.77|
> |Ours ($\lambda_1=20,\lambda_2=100$)|103.89|6.21|6.94|
> |Ours ($\lambda_1=20,\lambda_2=1000$)|85.72|10.16|11.87|
>
> While MMD with DINO serves as our primary quantitative evaluation metric for image quality, here we also report Fréchet Distance (FD) for generated images w/ and w/o DQA-Guidance.
> ||Avg. FD|Avg. FD Diff.|Max FD Diff.|
> |-|-|-|-|
> |Baseline|29.09|1.26|1.77|
> |Ours ($\lambda_1=20,\lambda_2=100$)|28.53|0.09|0.12|
> |Ours ($\lambda_1=20,\lambda_2=1000$)|26.27|0.29|0.44|
>
> Regarding the qualitative results in Figure 7, while some aspects of visual quality are subjective, the improvements are evident. For example, DQA-Guidance helps remove visual artifacts such as extra limbs (e.g., three hands), enhances image characteristics like color (e.g., from grayscale to natural color), and improves coherence between the prompt and the image (e.g., ensuring the nurse to be male along with a male prompt).
> ## Regarding Algorithm 1
> While Algorithm 1 may initially appear complex, its underlying logic is straightforward. It identifies subsets of generated images that most strongly increase or decrease the group-level discrepancy in perceived image quality, as measured by DQA.
>
> These subsets support a downstream diagnostic evaluation. Specifically, we use the fair/unfair subsets in classification as a data augmentation to assess whether the quality discrepancy is practically meaningful. We find that models trained on the unfair subset exhibit larger fairness gaps in downstream classification, while those trained on the fair subset show more equitable performance. This demonstrates that the discrepancy captured by the DQA is predictive of real fairness issues, thereby validating its utility as a diagnostic signal for metric reliability.
> ## Results in Table 1
> To show the validty of our experimental results in Table 1, we investigate the confidence interval of experimental results.
> ||Overall AUC|AVG($\Delta$AUC)|Max($\Delta$AUC)|
> |-|-|-|-|
> |Fair Subset| 53.91$\pm$ 0.26|6.18$\pm$ 0.36|15.65$\pm$ 2.25|
> |Unfair Subset| 54.32$\pm$ 0.24|6.83$\pm$ 0.39 |17.19$\pm$ 2.76 |
> ## Regarding the Vague Content
> Thank you for pointing this out. The sentence in Section 5 was intended as an introduction to the extension of DQA as a guidance term for diffusion models. We agree that the current phrasing may be vague and potentially confusing. We will revise this section to more clearly explain how DQA can be extended and applied in the context of guidance for diffusion models.

---

> > ### Comment · Reviewer_yKK1 · 2025-04-07
> >
> > Thank you for your response. Though the proposed image quality evaluation is new, the bias in the quality of the generated image has been widely explored. Why does the proposed method not compare with the existing methods [1,2,3,4] on the publicly available dataset rather than the self-constructed dataset? Thus, more experimental results are needed to support the superiority of the proposed DQA-Guidance.
> >
> > Here is a partial list of the references:
> > [1] Finetuning Text-to-Image Diffusion Models for Fairness
> > [2] INFELM: In-depth Fairness Evaluation of Large Text-To-Image Models
> > [3] Instructing Text-to-Image Generation Models on Fairness
> > [4] Unlocking Intrinsic Fairness in Stable Diffusion

---

> > > ### Author Response · Authors · 2025-04-07
> > >
> > > Thank you for raising this important point. However, the references cited by the reviewer—[1], [2], [3], and [4]—address a different type of bias than the one investigated in our paper. Specifically, these works focus on **distributional bias** in generative models, where the concern lies in the demographic distribution of generated images given a neutral prompt (e.g., aiming for a 50/50 balance between "male nurse" and "female nurse" for the prompt "a photo of a nurse"). In these studies, the evaluation metrics typically take the form of demographic parity, often expressed as "Ratio of Major Attribute" or "Rate of Female-Appearing" images.
> > >
> > > In contrast, our study investigates **quality bias**—disparities in the image quality of outputs across demographic groups when the group is explicitly specified in the prompt (e.g., higher-quality outputs for "a photo of a female nurse" than for "a photo of a male nurse"). We demonstrate that such bias not only exists in the generated outputs, but also that the commonly used evaluation metric, FID, fails to reliably detect these disparities.
> > >
> > > To address this dual issue, we propose DQA, which evaluates the reliability of image quality assessments using a self-constructed dataset that enables precise control over image quality for measuring reliability. We also introduce DQA-Guidance, which promotes fairness in image quality. To the best of our knowledge, prior work has not addressed fairness from the perspective of image quality, nor has it examined the reliability of image quality evaluation. For this reason, no suitable baseline methods were available for comparison, as briefly mentioned in our rebuttal.

---

### Official Review · Reviewer_4STN · 2025-03-17

**Overall Recommendation:** 2

**Summary:**

This paper introduces DQA, a novel scoring method designed to assess the reliability of image quality evaluation metrics, particularly in the context of generative models.  DQA aims to address the bias present in metrics like FID when evaluating image quality across different demographic groups. The core idea is to use carefully constructed, controlled datasets with comparable quality across groups and then measure the consistency of the image encoder. Furthermore, the paper proposes DQA-Guidance, a regularization technique applied during diffusion model sampling to mitigate quality disparities. The authors present empirical results demonstrating the effectiveness of DQA in identifying biased metrics and DQA-Guidance in improving fairness and overall image quality.

**Claims And Evidence:**

While the paper presents a compelling motivation and a novel approach, the evidence supporting the practical advantages of DQA over existing methods like FID is not entirely convincing. The claims regarding the superior fairness and reliability of DQA hinge on the construction of the controlled datasets.  It's unclear how robust these datasets are to different types of quality degradation and whether they truly capture the nuances of real-world image quality variations across demographic groups.  The gains achieved with DQA-Guidance, while present, appear somewhat marginal and their significance needs further validation.

**Essential References Not Discussed:**

NAN

**Experimental Designs Or Analyses:**

The experimental designs are generally well-structured, but the analysis could be more in-depth. For instance, a more detailed ablation study exploring the impact of different components of DQA-Guidance (e.g., the regularization parameters) would be valuable.  The paper should also provide more insights into the computational cost associated with DQA and DQA-Guidance compared to standard FID-based evaluations and sampling.

**Methods And Evaluation Criteria:**

The proposed methods are well-defined and logically sound.  However, the evaluation criteria rely heavily on the artificially constructed datasets.  The paper would benefit from a more rigorous evaluation using real-world datasets with inherent demographic biases, even if it's challenging to establish ground truth quality.  The choice of MMD as the distance metric is justified, but exploring alternative distance metrics and analyzing their impact on DQA scores would strengthen the analysis.

**Other Comments Or Suggestions:**

The authors should clearly articulate the assumptions underlying the construction of the controlled datasets.
The paper should include a discussion of the potential limitations of DQA when applied to datasets with complex or unknown demographic biases.

**Other Strengths And Weaknesses:**

Strengths: The paper addresses an important and timely problem: fairness in image generation. The DQA metric provides a useful tool for evaluating the reliability of image encoders. The DQA-Guidance method offers a practical way to mitigate quality biases in diffusion models without retraining.
Weaknesses: The reliance on artificially constructed datasets limits the generalizability of the findings. The experimental results are not entirely convincing, and the computational cost of DQA is not adequately addressed. The paper could benefit from a more in-depth analysis of the limitations of the proposed approach.

**Questions For Authors:**

How do you ensure that the artificially constructed datasets used for DQA accurately reflect real-world image quality variations across different demographic groups? How sensitive are the DQA scores to the specific choices made in constructing these datasets (e.g., the types and magnitudes of the quality degradations)? A response demonstrating the robustness of the dataset construction would strengthen my confidence in the reliability of DQA.
Can you provide a more detailed analysis of the computational overhead associated with DQA and DQA-Guidance compared to standard FID and diffusion model sampling? Quantifying the extra cost would help assess the practicality of the proposed methods.
What are the limitations of DQA-Guidance? Under what circumstances might it fail to improve or even worsen the fairness of the generated images? Addressing this limitation would show a more balanced perspective.

**Relation To Broader Scientific Literature:**

The paper builds upon a foundation of existing work on fairness in machine learning and image quality assessment.  It correctly identifies the limitations of FID in the context of demographic biases and proposes a novel approach to address these limitations.  The related work section is comprehensive.

**Theoretical Claims:**

NAN

---

> ### Author Rebuttal · Authors · 2025-04-01
>
> ## Adjustment in the Controlled Dataset
> The degradations we introduce are well-established in diffusion-based generative models literature.
> - **Weak Classifier-Free Guidance (CFG)**
>   In CFG, using weak guidance simulates a scenario where the generated image loses coherence with the prompt.
> - **Fewer Inference Steps**
>   As noted in [1], images generated with fewer diffusion steps typically exhibit lower visual quality by leaving more residual noise and artifacts.
> - **Stronger Initial Noise**
>   In our method, we use in-painting to construct the controlled dataset. We first generate images from text prompts and then modify them to reflect different attributes to maintain contextual consistency. In in-painting, a stronger noise level preserves more of the original image. As a result, the model struggles to apply the desired attribute modification, leading to poor coherence with the target attribute.
> - **No Refiner**
>   The SDXL paper explicitly states that the use of a refiner network improves visual quality, meaning removing the refiner leads to a noticeable degradation in image quality.
>
> Each of these modifications is grounded in existing literature, ensuring that the controlled degradations reflect realistic variations in generation quality.
>     [1] Kim et. al., 2024, Model-Agnostic Human Preference Inversion in Diffusion Models
>
> ## Hyperparameter Sensitivity
> As shown in the figure at the following link:
> https://drive.google.com/file/d/1Ot1FkMuPYmb0-6vFZZ5vUHmtcw6xgGAq/view?usp=sharing
> we investigate the impact of hyperparameter variation in the controlled dataset by adjusting the guidance scale in CFG. The right subfigure shows a clear degradation in the generated images as controlled.
>
> In the main paper, we argue that a lower average DQA across degraded images indicates higher reliability of an evaluation metric. Although this conclusion remains consistent (DINO-RN50 as the most reliable), this additional analysis highlights that robustness to degradation could serve as another criterion for evaluating the reliability of an image encoder.
>
> We will include this analysis in the revised version to better emphasize the desired characteristics of reliablilty for quality evaluation.
>
> ## Analysis for Experimental Results
> Please see the rebuttal for Reviewer yKK1.
> ## Regarding Ablation Study
> In Figure 6, the left subfigure shows the performance trend when varying $\lambda_1$ while keeping $\lambda_2$ fixed, and the right subfigure shows the trend when varying $\lambda_2$ with $\lambda_1$ held constant. Therefore, the experimental results in Figure 6 already serve as an ablation study, demonstrating the individual effects of each component in the DQA-Guidance.
> ## Cost for DQA Evaluation
> Since DQA is a **reliability score** for evaluation metrics, its cost is not directly comparable to performance metrics such as FID. However, to clarify the computational requirements: DQA involves three quality evaluations, two for the subgroup qualities (numerator) and one for the overall quality (denominator).
> Importantly, DQA needs to be computed only once for each encoder to determine the reliability for evaluation. Once a reliable score is identified, there is no need to recompute DQA during future model evaluations.
> On the other hand, MMD with DINO is our primary quantitative evaluation metric. Notably, MMD is reported to be approximately 1,000 times faster than the Fréchet Distance used in FID, making it a more efficient choice for large-scale or repeated evaluations.
> ## Cost for DQA-Guidance
> We observe the increase in memory usage. DQA-Guidance utilizes an additional image encoder along with gradient computation for the guidance term. As a result, memory usage increased from 10,124MB to 18,626MB due to the gradient computation.
> ## Potential Limitation
> ### Computational Cost
> As shown above, DQA-Guidance introduces additional computational cost. However, since ours is the first work to demonstrate that quality bias-based guidance can steer diffusion models toward fairer outputs, it opens a promising direction for fairness-aware guidance. Future work can further explore cost-efficient implementations of such approaches.
> ### Lack of Real Reference Dataset
> Although the controlled dataset contains high-quality images, they may not fully align with human-perceived realism or quality standards. A promising future direction is to develop human-validated reference datasets, where quality judgments are collected through perceptual surveys. This would enhance the validity of DQA as a reliability indicator and offer a more robust benchmark for auditing image evaluation metrics.
> ### Possibility of Overfitting
> DQA-Guidance leverages group-specific references. However, if the reference set is narrow or unrepresentative, the model might overfit to a particular visual style, reducing diversity or realism. Constructing more diverse and well-controlled reference dataset would improve the generalizability of DQA-Guidance.

---

### Decision · Program_Chairs · 2025-05-01

**Decision:**

Reject

**Comment:**

Two weak rejects, one weak accept, and one accept (though short and unreliable review).

Given the divergence, AC read the paper, reviews, rebuttals, and additional discussion carefully.

For one of the two weak rejects, some discussion followed, though without explicit conclusion (the reviewers didn't continue the discussion). For the other weak reject, there was no interaction at all (the reviewer didn't respond).

There were some main critiques raised by the reviewers:
- Lack of novelty of the approach
- Need for well-balanced reference data for evaluation
- Marginal improvement with DQA guidance
- Presentation issues

Here's the conclusion from the AC:
- Novelty: AC acknowledges that the problem is novel. One of the weak reject reviewers has misunderstood that the novelty is shaded by the generation fairness papers that have studied the demographic representations in generations. However, the current submission contributes to the measurement of generation qualities across different demographics.
- Yes, the method requires well-balanced reference data for evaluation. This is a clear practical limitation.
- Marginal improvement is not a problem from AC's perspective, as it's only a secondary contribution.
- There are several presentation issues, along with other concerns that AC shares with the reviewers, including the lack of readability in both the figures and the text throughout the paper.

Given this, AC believes that the paper should still be accepted, acknowledging its first contribution towards the problem of demographic biases in generation quality. The acceptance is conditional upon the promise that the authors greatly improve the presentation quality and make the limitations described in the rebuttal to 4STN clear in the manuscript.

PC intervention: this paper was quite borderline, and there is no such thing as conditional acceptance at ICML, so the recommendation was moved to "revise and resubmit" at the next conference so that the revision can be properly checked by reviewers.